

# Experimental study of the aerosol impact on fog microphysics.

M. Mazoyer[1], F. Burnet[1], G. C. Roberts[1], M. Haeffelin[2], J.-C. Dupont[2], and T. Elias[3]

[1]Meteo-France/CNRS - CNRM/GAME, Toulouse, France
[2]Institut Pierre Simon Laplace/UVSQ, Palaiseau, France
[3]HYGEOS, Lille, France

*Correspondence to:* M. Mazoyer (marie.mazoyer@meteo.fr)

**Abstract.**

Comprehensive field campaigns dedicated to fog life cycle observation were conducted during the winters of 2010-2013 at the SIRTA observatory in the suburb of Paris. In order to document their properties, in situ microphysical measurements collected during 23 fog events are examined here. They reveal large variability in number, concentration and size of both
aerosol background before the fog onset and fog droplets according to the different cases. The objective of this paper is to evaluate the impact of aerosol particles on the fog microphysics.

To derive an accurate estimation of the actual activated fog droplet number concentration $Nact$, we determine the hygroscopicity parameter $\kappa$, the dry and the wet critical diameter and the critical supersaturation for each case by using an iterative procedure based on the $\kappa$-Köhler theory that combines cloud condensation nuclei (CCN), dry particle and droplet size dis-
tribution measurements. Resulting values of $\kappa = 0.17 \pm 0.05$ were found typical of continental aerosols. Our study reveals low values of the derived critical supersaturation with a median of 0.043 % and large values for both wet and dry activation diameters. Consequently, the corresponding $Nact$ values are low with median concentrations of 53.5 $cm^{-3}$ and 111 $cm^{-3}$ within the percentile 75th.

No detectable trend between the concentration of aerosol particles with diameter > 200 $nm$ and $Nact$ was observed. In
contrast the CCN data at 0.1 % supersaturation exhibits a strong correlation with these aerosol concentrations. We therefore conclude that the actual supersaturations reached during these fog episodes are too low and no simultaneous increase of aerosols > 200 $nm$ and droplet concentrations can be observed. Moreover our analysis suggests that a high aerosol loading limits the supersaturation values. It is also found that the activated fraction mainly depends on the aerosol size while the hygroscopicity appears to be of a secondary importance.

Although radiative fogs are usually associated with higher aerosol loading rather than to stratus lowering events, our analysis reveals that the activated particle concentrations at the beginning of the event are similar for both types of fog. However the evolution of the droplet concentration during the fog life cycle shows significant differences between both types of fog.

## 1   Introduction

As they reduce visiblity, fog events strongly perturb the aviation, marine and land transportation. Furthemore Gultepe et al.
(2009) pointed out the extremely high level of human losses and financial cost related to fog and low visibility events. Moreover



present numerical weather prediction models are usually unable to predict the exact location and time evolution of the life cycle of a fog layer (Zhou and Ferrier, 2008; Van der Velde et al., 2010; Bergot, 2013).

Indeed, the occurence and development of fog result from the nonlinear interaction of competing radiative, thermodynamic, microphysical and dynamical processes (see the review of Gultepe et al. (2007b)). To better understand the relationship between such these processes, field campaigns are carried out at the Instrumented Site for Atmospheric Remote Sensing Research (SIRTA) in the suburb of Paris (Haeffelin et al., 2010). From October 2010 to March 2013 a suite of active and passive remote sensing and in situ sensors were deployed to provide state of the art observations of wintertime fog events. Dupont et al. (2015) analysed 117 fog events induced by radiative cooling (53 events) and by stratus lowering (64 events) to provide typical values of thermo-dynamical and radiative variables for the fog formation, the mature and dissipation phases, and the range of values that criticals parameter have must reach for fog and quasi-fog to form.

One of the strong points of the instrument set-up was the in situ measurements of particle size distribution. Burnet et al. (2012) have shown that, in term of droplet number concentration and effective diameter, the microphysical properties of fog events sampled at SIRTA present a large variability. Elias et al. (2015) examined the data collected during Nov 2011 and assessed the contribution of hydrated aerosols that led to the extinction of the visible radiation in the mist-fog-mist cycle. Hammer et al. (2014) focused on the activation properties of 17 developed fog events observed during winter 2012/2013 by measuring total and interstitial dry particle behind two different inlet systems. Stolaki et al. (2015) analysed the sensitivity of fog to aerosols through their impacts on fog droplets. Their numerical sensitivity study shows that the characteristics of fog are strongly influenced by the aerosols.

Here we examined in situ microphysical measurements collected during the 3 winters campaigns to investigate the impact of aerosols on the fog microphysics.

Water droplets are formed by heterogeneous nucleation of aerosol particles when the relative humidity (RH) exceeds 100 %. The Köhler theory (Köhler, 1936) forms the basis of our understanding of cloud droplet formation. The ability of aerosol particles to act as CCN (Cloud Condensation Nuclei) depends largely of their size, composition and the phase state. The number size distribution of the droplets activated during the cloud or fog formation phase depends on the supersaturations values reached by the air mass (Pruppacher et al., 1998). Those particles having a critical supersaturation below the maximum value are activated and will then further grow by water vapour diffusion as long as the RH remains high enough while the others particles remain at their equilibrium diameter.

As a general rule in cloud physics, for a given cloud type, the more aerosol particles, the more droplets (Ramanathan et al., 2001). Hudson (1980) found systematic differences in the fog microstructure between fogs formed in maritime, continental and urban air masses with fog condensation nuclei and an increase in drops concentration, respectively. On the other hand, numerical study of Bott (1991) shows that an increase in aerosol particles leads to a decrease of the supersaturation that in turn decreases the activated droplet number. But recent numerical simulations exhibit a strong positive correlation between aerosol and droplet concentrations (Zhang et al., 2014; Stolaki et al., 2015).



Moreover because of the weak supersaturation inside the fog layers (Hudson, 1980; Pandis et al., 1990; Svenningsson et al., 1992; Hammer et al., 2014) the separation between activated fog droplets and non activated particles so-called hydrated aerosols is not explicit (Frank et al., 1998). However the microphysics in NWP (Numerical Weather Prediction) models take account activated droplets only. In addition, a two-moment microphysical cloud scheme has been recently developed for low saturation clouds (Thouron et al., 2012) and needs to be tested against observations. Prior validating a numerical model, it is thus essential that the total droplet number concentration be precisely evaluated.

The objective of this study is thus to derive typical fog droplet concentrations in semi-urban condition to investigate the impact of aerosols on the fog microphysics. To derive accurate estimations we determine the hygroscopicity parameter, the dry and wet critical diameters, and the critical supersaturation by using an iterative procedure based on the $\kappa$-Köhler theory that combines CCN measurements, dry particle size distribution and composite wet particle distribution at ambient humidity. Data and method are described in Sect. 2. and 3., respectively. Results are presented in Sect. 4 with first the statistics of fog activation properties of the 23 fog events analysed in this study. The link between aerosol particles and fog droplets are examined in Sect. 4.2, and the impact of CCN concentration on fog microstructure is discussed in Sect. 4.3. Conclusions are finally given in Sect. 5.

## 2  The dataset

### 2.1  Instrumentation

Data presented here were collected at the SIRTA observatory in the framework of the ParisFog field campaigns (Haeffelin et al., 2010). During the winters 2010 to 2013, specific instrumentation were deployed for the PreViBOSS project (Elias et al., 2012) to provide continuous observation of aerosol and fog microphysics. The experimental set-up was already presented in Burnet et al. (2012); Hammer et al. (2014); Elias et al. (2015); Dupont et al. (2015). The instruments used in this study are listed in Table 1.

Particle size distribution at ambient humidity is derived from a combination of two optical spectrometers: the WELAS-2000 (Palas Gmbh, Karlsruhe, Germany) and the FM-100 (Droplet Measurement Techonologies Inn., Boulder, CO, U.S.A.). The WELAS-2000 (hereafter referred as WELAS) provides particle size spectrum between 0.4 and 40 $\mu$m in diameter. However the detection efficiency decreases drastically below $\sim 1$ $\mu$m (Heim et al., 2008; Elias et al., 2015) resulting in a strong under-estimation of the concentration of the submicronic particles. Hammer et al. (2014) choose to consider only data with diameter larger than 1.4 $\mu$m. Statistics over the whole data set reveals that the most frequent mode diameter of the WELAS size distribution is 0.96 $\mu$m. Thus we choose to use this value as the lowest threshold and only measurements for bin diameter larger than 0.96 $\mu$m will be considered in this study. Note that the activated diameter in fog is expected to be larger that 1 $\mu$m, thus this instrumental bias will not affect the results presented here. The sampling time period was fixed to 5 minutes as a compromise between time resolution and statistical significance of the measurements. Indeed this corresponds to a volume of air of 6.40 $cm^{-3}$ per sample.



The FM-100 provides 1 Hz droplet size distribution from 2 to 50 $\mu$m in diameter. Thus there is a large overlapping range with the WELAS measurements. However these distributions overlap each other at a diameter which fluctuates between 5 to 9 $\mu$m and comparisons reveal a high discrepancy between both these probes with a large underestimation by the FM-100 for particle less than about 5 $\mu$m but conversely a large underestimation by the WELAS for droplets larger than 10 $\mu$m (Burnet et al., 2012;

Elias et al., 2015). This is illustrated in Fig. 1 which shows the size distributions measured by the WELAS (black) and FM-100 (cyan) for two fog cases with contrasting properties. Figure 2 shows comparisons of the particle number concentration values of the FM-100 vs the integrated WELAS measurement over the same diameter range, for the four first bins of the FM-100. The default manufacturer's first four bins of the FM-100 are [2-4], [4-6], [6-8] and [8-10] $\mu$m which correspond to 9, 7, 4 and 3 class bins of the WELAS, respectively. Data points are 5 min average that represents 2851 samples for the whole data

set. Figure 2 confirms that the FM-100 strongly underestimates the particle counts in the first size bin and that the WELAS underestimates the concentration of particles larger than 8 $\mu$m. The two instruments do not match perfectly each other over particles from between 4 and 8 $\mu$m, which reflects the large fluctuations of the crossing diameter over this range. To derive a composite size distribution, Elias et al. (2015) choose a constant value of 7 $\mu$m. To take into account such a variability, we compute here the composite size distribution by using WELAS data up to 6 $\mu$m, FM-100 data above 8 $\mu$m and the average

of both from 6 to 8 $\mu$m. Results are illustrated in Fig. 1 with red segments corresponding to the junction between WELAS and FM-100 distribution. This procedure allows us to take advantage of the finer resolution of the WELAS for the smaller particles and also to reduce uncertainties of the FM-100 due to Mie ambiguities as described by Spiegel et al. (2012) since our FM-100 was using the default manufacturer's bin threshold. Indeed the droplet concentration, in some size classes can be overestimated or underestimated, a few droplets from adjacent classes can be included or few some can be counted in an

adjacent class (Gonser et al., 2012).

Both instruments are located on a scaffolding at about 2.5 m high, close to a PVM-100 from Gerber Scientific Inc. used as a reference for the LWC measurements. Measurements of visibility and its vertical evolution are given by two Degreanne diffusometers (DF20+ and DF320) located at 4 m and 18 m above ground, respectively.

Aerosol particles measurements are performed by instruments placed in a shelter. The sampled air mass enters through an

aerodynamic size discriminator PM 2.5 inlet and a dryer which reduces the relative humidity to less than 50 %. A scanning mobility particle sizer (SMPS) which provides the dry aerosol particle number size spectrum, consists of a differential mobility analyzer (DMA; TSI 3071) which selects particles from 10.6 to 496 nm and of a condensation particle counter (TSI CPC 3022). Another CPC (TSI 3025) measures the total particle number concentration from 2.5 nm to 2.5 $\mu$m. Finally a continuous flow streamwise thermal gradient CCN chamber (Roberts and Nenes, 2005) is used to measure the CCN number concentration

at 5 different supersaturations from 0.1 % to 0.5 %, by step of 0.1 %.

## 2.2 Selected fog cases and aerosol properties

During the three wintertime campaigns of 2010-2013, Dupont et al. (2015) report the occurrence of 117 fog events. However due to instrument failures and technical difficulties in operating the whole set of instrument on a 24/7 mode, 42 events were





sampled simultaneously with both the WELAS and the FM-100, and only a subset of 23 cases were also sampled with both the CCNC and the SMPS. They are listed in Table 2 with their classification type, RAD for radiative cooling fog and STL for stratus lowering ones as determined by the scheme of Tardif (2007), and their vertical development based on the comparison of both difusometers : a developed fog produces low visibility condition simultaneously at 4 m and 18 m while a thin fog produces low visibility condition at 4 m only (Elias et al., 2009; Dupont et al., 2015). About the same proportion (40 %) of RAD and STL fog events occured at SIRTA site Haeffelin et al. (2010); Dupont et al. (2015). Of the 23 fog events analysed here, 13 are radiation fog and 10 stratus lowering fog, 19 are developed and 4 thin.

To characterize the aerosol background prior to a fog event, statistics of the total number aerosol particle concentration, $Na$, as measured by the SMPS are computed over the last hour before the beginning of the fog event. Median, 25th and 75th percentiles are reported in Table 2. $Na$ values range from $\approx 2000\ cm^{-3}$ to $\approx 20000\ cm^{-3}$. Consistently with (Haeffelin et al 2010) the smallest values are observed in westerly flow conditions while highest values are associated mostly with an easterly flow when SIRTA is exposed to continental conditions. Indeed SIRTA is located 25 km south-west of Paris in a semi-urban environment, composed of agricultural fields, wooded areas, housing and industrial developments (Haeffelin et al., 2005) and is exposed to air mass charged with pollution originating from regional background according to Crippa et al. (2013).

These various conditions are reflected on the statistics that reveal a large case to case variability. This is illustrated in Fig. 3 that shows the scatter-plot of the droplet number concentration derived by the FM-100, $N_{FM}$, (table 2) as function of $Na$ for the 23 fog events. Symbols depend on the fog type : blue and red colors for STL and RAD fogs, respectively, and open and solid symbols for thin and developed fogs, respectively. For STL fogs, $Na$ ranges from 2000 to 11000 $cm^{-3}$ with a median value of 4340 $cm^{-3}$ and 25th and 75th percentiles of 2833 and 6942 $cm^{-3}$, respectively, while for RAD cases, $Na$ are spread over a larger range from 3000 to 20000 $cm^{-3}$ with a median value of 8822 $cm^{-3}$ and 25th and 75th percentiles of 5719 and 13094, respectively. Thus over this three year campaign it appears that at SIRTA STL fogs are associated with lower aerosols loading than RAD ones, as already reported by Elias et al. (2015) for the month of November 2011.

There is also a significant difference between a STL and RAD fog with respect to $N_{FM}$. Despite that the maximum value is reached for a STL fog (147 $cm^{-3}$ for the case f7) $N_{FM}$ for a STL fog is generally lower with median values between 38 and 83 $cm^{-3}$ and 25th-75th percentiles of 31-63 and 53-118 $cm^{-3}$, for STL and RAD, respectively. Moreover, for a RAD fog, thin cases exhibit higher values of $N_{FM}$ with median and 25th-75th percentiles of 97 and 73-131 $cm^{-3}$ compared to 64 and 49-81 $cm^{-3}$ for a developed fog. Therefore it appears that a radiative fog is also associated with higher droplet concentrations over the range [2-50] $\mu$m as measured by the FM-100. One can note that these values are rather low for continental conditions.

As shown in Fig. 3 the general trend point towards a slight increase in the numer of droplets with the aerosol loading but the scatter svery large. For instance, for a droplet concentration of 60 $cm^{-3}$, a value close to the median value for all cases, aerosol background values as low as 3500 and as large as 20000 $cm^{-3}$ are observed.

In cloud physics the connection between an increase in aerosol particles and an increase in the cloud droplet number concentration (CDNC) has been supported by many in situ observations (Twohy et al., 2005; Lu et al., 2007; Levin and Brenguier,



2009) and even if the discrepancy is large in the compilation of the diverse results (Ramanathan et al., 2001) the general trend is much more pronounced to what is presently observed in Fig. 3. However, CDNC is derived here from FM-100 measurements only. In warm clouds the supersaturation at the cloud base is high enough to activate droplets whose size will increase by water vapour condensation during the ascent of the air parcel. The spectrum then gets narrower because the growth rate

of a droplet is inversely proportional to its size, while the remaining interstitial particles keep their own equilibrium diameter at 100 % relative humidity. Higher up in the cloud the spectrum is widened by different processes such as turbulent mixing and collision-coalescence and new CCNs can also be activated. Droplet spectra vary considerably in space and in time but the general trend is that the activated droplet population is clearly separated from the interstitial non-activated particles. See for example measurements into stratocumulus cloud as reported in Martin et al. (1994); Brenguier et al. (2011); Ditas et al. (2012).

Measurements by optical counters such as FSSP or CDP over the range [about 2-50 $\mu$m] are considered to provide an accurate estimation of the droplet size distribution.

In contrast, supersaturation in fog is much lower and, as already pointed out by Hudson (1980), in continental air the droplet number size distribution does not exhibit a clear separation. This is clearly illustrated in Fig. 1. To derive an accurate estimate of the CDNC it is thus essential to determine carefully the wet activation diameter in order to integrate the composite size

distribution derived from both WELAS and FM-100 measurements. The method used to discriminate the droplets from the hydrated aerosol particles using kappa-Köhler theory is described in the next section.

## 3 Methods

There are different methods to separate fog droplets from non-activated aerosol particles. A fixed value can be used as a rough estimate as described in previous studies by Noone et al. (1992); Hoag et al. (1999); Elias et al. (2009) among others,

with thresholds ranging from 2.5 to 5 $\mu$m. To take into account case to case variability, Elias et al. (2015) used WELAS measurements. The volume size distribution is fitted with two log-normal distributions and the transition diameter is defined as the intersection between them.

Hammer et al. (2014) investigated the activation properties by measuring the total and interstitial dry particle number size distribution behind two different inlet systems, and with WELAS and CCN measurements. They compared two methods one

by fitting surface distributions of SMPS+WELAS measurements similar to Elias et al (2014) and one by retrieving the dry activation diameter from the difference between interstitial and total particle size distribution measurements.

In this study we use a different approach based on an iterative procedure that combines dry particle distribution from SMPS and composite wet particle size distribution derived from both WELAS and FM-100 measurements. The CCN measurements are also used to derive the hygroscopicity parameter ($\kappa$) needed to link the dry and wet activation diameter ($D_d$ and $D_w$) with

the critical supersaturation peak ($SS_{peak}$) using the $\kappa$-Köhler theory.





### 3.1 Kappa-Köhler theory

The Köhler theory (Köhler, 1936) expresses the equilibrium saturation vapor pressure over a solution droplet considering the solute effect (Raoult) and water surface tension effect (Kelvin). Accurate information is needed on the particle dry diameter and about the chemistry to determine wether or not it can act as a CCN. Recently Petters and Kreidenweis (2007) developed a method, named $\kappa$-Köhler, to describe the relationship between particle dry diameter and CCN using a single hygroscopic parameter, $\kappa$. This method allows to study the activation process without considering aerosols complex chemistry (McFiggans et al., 2006). The formulation of the method is expressed in Eq.(1):

$$S(D) = \frac{D^3 - D_d^3}{D^3 - D_d^3(1-\kappa)} exp(\frac{4\sigma_{s/a}M_w}{RT\rho_w D})$$ (1)

where S is the saturation ratio over a solution droplet, D is the droplet diameter, $D_d$ is the dry droplet diameter, $\rho_w$ is the density of water, $M_w$ is the molar mass of water, $\sigma_{s/a}$ is the surface tension of the solution air/interface (of pure water here), R is the universal gas constant, T is the temperature and $\kappa$ is the hygroscopicity parameter. $\kappa$ represents a quantitative measure of aerosol particles water uptake characteristics and CCN activity. The critical supersaturation $SS_c$ and critical wet activation diameter $D_w$ correspond to values at maximum of supersaturation. They are linked to a couple ($D_d$, $\kappa$) through the following relation :

$$\kappa = \frac{4A^3}{27D_d^3 ln^2 SS_c} \qquad A = \frac{4\sigma_{s/a}M_w}{RT\rho_w}$$ (2)

$SS_{peak}$ represents the maximum supersaturation that the air mass experienced for a sufficiently long time (Hammer et al., 2014), all particles whose $SS_c$ is less than $SS_{peak}$ are activated and will further grow by water vapour condensation as long as RH remains high enough while other particles remain stable at their equilibrium diameter at the actual RH value.

### 3.2 Method to determine fog activation properties

The iterative process used to derive fog activation properties is illustrated in Fig. 4. The idea is to match the concentration of CCN, $N_{ccn}$, calculated as the integral from $D_d$ of the dry aerosol particle distribution measured by the SMPS with the fog droplet number concentration $N_d$, calculated as the integral from $D_w$ of the composite wet particle size distribution derived from both WELAS and FM-100 measurements. Indeed it can exist only one trio $N_{ccn}$, $D_w$ and $D_d$ that are linked to $\kappa$ and $SS_{peak}$ through Eq.(2).

To determine the dry activation diameter we suppose that all particles with a dry diameter larger than $D_d$ are activated, which means that aerosols are internally mixed. Indeed our instrumental system allows us to determine the CCN concentration by supersaturation but not by size. Jurányi et al. (2013) showed that aerosol particles are often externally mixed at SIRTA but they also noticed that the way of treatment of mixing states does not significantly influence the predicted CCN. We also suppose that the concentration of aerosol particle larger than 496 nm can be neglected.

The air mass is sampled trough a PM2.5 head and is dried to RH < 50 % before entering the SMPS and the CCN chamber. Hence during a fog event activated aerosols larger than 2.5 $\mu$m are missed. Aerosol properties of the air mass must thus be



characterized before the occurence of a fog event. But the aerosol particle distribution changes continuously according to the boundary layer evolution and aerosol sources (wood burning, road traffic, ...). This is illustrated in Fig. 5 that illustrates the time series of measurements for the fog event f6. There is a large variability along the diurnal cycle. To characterize the air mass just before the fog event, we average data recorded during the last hour before the fog onset. This corresponding time interval is delimitated by red segments in Fig. 5.

CCN chamber measurements are used to derive $\kappa$ by using Eq.2 knowing $SS_c$ and $D_d$. The chamber supplies the concentration of activated aerosol particles $N_{ccn}$ at five supersaturations from 0.1 % to 0.5 %. Scanning takes about 20 minutes.

Therefore 3 scans can be made during the 1 hour interval thus provides meaningful estimation of the aerosol hygroscopicity. But supersaturation values in the chamber are greater than 0.1 % and it is well known that $SS_{peak}$ in fog is generally lower than 0.1 % (Hammer et al., 2014; Ming and Russell, 2004; Svenningsson et al., 1992; Hudson, 1980). Therefore assumptions must be made to extrapolate $\kappa$ at lower supersaturation. This is illustrated in Fig. 6 that shows statistics of $\kappa$ as a function of supersaturation for the selected time period of Fig. 5. Data points correspond to calculations with CCN data at each supersat-

uration and the SMPS aerosol size distribution averaged over the one hour time interval. Solid and dashed lines superimposed to the data points correspond to mean and mean +/- one standard deviation values, respectively. Two extrema of $\kappa$ are defined. $\kappa_{inf}$, which is computed as the mean value of $\kappa$ at 0.1 % minus standard deviation, corresponds to the lowest expected value. $\kappa_{sup}$ is calculated as the linear extrapolation of the mean $\kappa$ at 0 % plus one standard deviation and corresponds to the highest expected value. Both values are indicated in Fig. 6 by blue and red dots, respectively. These two extreme values of $\kappa$ will then

provide extreme values of activation properties for a given dry activation diameter.

Once droplets have been activated, their size distribution evolves during the fog life cycle. They can grow by water vapour diffusion or by coalescence with other droplets due to gravitational and turbulent motions and they can also evaporate in case of mixing with clear air or changes in temperature. Following Noone et al. (1992), the mixing with clear air has similar

consequences as of external mixture for aerosols, that means that some particles can deactivate and their diameter can be under the critical wet diameter because of evaporation. In convective clouds, the activation occurs mainly within the first tenth of meters above the cloud base and it is thus possible to directly sample the resulting droplet spectra with instrumented aircraft. Whereas, in case of fog, the activation first occurs at the fog onset, the ensuing vertical development of the fog layer depends on many processes among which the radiative cooling at the fog top plays a key role. The fog onset is generally defined by a

drop of visibility below the 1 km threshold, but this is still subject to debate (see discussion in Elias et al. (2015)). Time series show indeed various case to case CDNC evolution during the first fog hour of the fog event, with an instantaneous formation, or conversely, a slower increase towards a stable value as illustrated in Fig. 5. To estimate the most representative fog droplet size distribution of the activated distribution, we average the composite wet particle size distribution derived from both WELAS and FM-100 measurements over a time interval from the beginning of the fog event during which the CDNC reaches a stable

value for a sufficient long time. On average, this time period is selected from 30 min to 1 hour after the fog beginning. It is delimited by blue segments in Fig 5. We therefore also assume that local measurements at 2 m height are representative of the fog layer. Furthermore we assume that the concentration of droplets larger than 50 $\mu$m can be neglected.





For each fog event the activation properties are determined as follows : $N_{ccn}$ is set corresponding to a value of $D_d$. $D_w$ and $SS_c$ are further calculated by numerically searching for the maximum of Eq. 1 for a given value of $\kappa$. Then the integral of the droplet distribution from $D_w$ provides the droplet concentration $N_d$. Iterations are made on $N_{ccn}$ until $N_{ccn}$ equals $N_d$ : this is the activated aerosol number concentration for this case hereafter referred as $N_{act}$. Two sets of iteration are made with the two $\kappa$ values and the average particle size distribution. To take into account the size distribution variability within the selected time period two other sets of iteration are made by using the average size distribution +/- one standard deviation, respectively, and with the $\kappa$ value maximizing the scatter.

Fig. 7 illustrates the impact of such a variability and shows the results for two fog cases of Fig. 1. The average aerosol dry size distribution measured by the SMPS recalculated at 100 % RH with the $\kappa_{inf}$ and $\kappa_{sup}$ values (blue and purple lines, respectively) are superimposed to the composite fog droplet number size distribution. Vertical segments indicate the corresponding $D_w$ diameters. These values are 3.37 and 5.45 $\mu$m corresponding to $N_{act}$ of 91 and 61 $cm^{-3}$, respectively for the f6 case (Fig. 7-a); and 2.15 and 3.08 $\mu$m which correspond to $N_{act}$ of 116 and 86 $cm^{-3}$, respectively for the f20 case (Fig. 7-b). The dashed ones correspond to the two calculations which produce extreme values and thus delimit the range of possible values. If the spectrum remains constant as in Fig. 7-a, these values are rather identical. But if the shape of the spectrum evolves during the one hour time period, the range of possible values increases as in Fig. 7-b with extreme values of 1.4 and 4.75 $\mu$m corresponding to $N_{act}$ of 266 and 21 $cm^{-3}$, respectively, leading to an uncertainty of a factor 10 for $N_{act}$. There is a good agreement between both distributions on the overlap area corresponding to hydrated (non-activated) particles. This indicates that the hygroscopic grow of all dry particles measured before the fog event with the derived $\kappa$ values is consistent with the ambient measurements at the beginning of the fog. Moreover this confirms that our method provides a satisfactory estimation of $\kappa$ and that the selected time periods are adequate to estimate the activation properties.

## 4 Results

### 4.1 Fog activation properties

Table 3 presents results for the 23 fog events with mean values and uncertainty intervals for each parameter $\kappa$, $D_d$, $D_w$, $SS_{peak}$ and $N_{act}$. The mean value is the average between the two values obtained from the above procedure with the average wet composite size distribution and the two $\kappa$ values. The uncertainty interval is determined by the extreme values resulting from the average wet composite size distribution minus (plus) one standard deviation and the lowest (highest) $\kappa$ value, respectively.

Values of $\kappa_{inf}$ and $\kappa_{sup}$ are relatively close. Indeed the relative standard deviation to the average value $\kappa_{mean}$ ranges between 9 and 25 % except for the case f14 which reaches 35 %. It follows that $\kappa_{mean}$ which corresponds roughly to $\kappa$ value at SS≈0.05 %, is representative of the $\kappa$ value at the actual supersaturation. As a result $\kappa_{mean}$ values range from 0.09 to 0.3 with a median of 0.17. They are in agreement with those determined by Hammer et al. (2014) who found $\kappa$ between 0.06 and 0.27 centred at 0.14 (SS ≤ 0.11 %) and Jurányi et al. (2013) that reported $\kappa$ between 0.08 and 0.24 (SS=0.1-1 %) at the same



site. They are slightly lower than usual for continental aerosols : Andreae and Rosenfeld (2008) for example suggested to use $\kappa = 0.3 \pm 0.1$. As already pointed out by Hammer et al. (2014) local emissions of road traffic and residential wood burning as likely responsible to these low values of $\kappa$.

The mean dry activation diameter values, 0.39 $\mu$m for the median and 0.35-0.43 $\mu$m for the 25th-75th percentiles, is rather high, which indicates that only largest aerosol particles are activated in agreement with Noone et al. (1992), Ming and Russell (2004) and Hammer et al. (2014) results. Concerning the mean wet activation diameter, $Dw_{50th} = 3.79$ $\mu$m with 3.03-4.67 $\mu$m for the 25th-75th percentiles. These values are slightly higher than values reported by Hammer et al. (2014), $Dw_{50th} = 2.6$ $\mu$m, but consistent with Elias et al. (2015) results $Dw = 4 \pm 1.1$ $\mu$m who determined it from a November 2011 dataset as the intersection between the two log-normal distributions fitting the particle volume distribution measured by the WELAS. Such a gap between hydrated and activated particles can be seen in figures 7-a) and b) and it corresponds fairly well to the average wet activation diameter derived for such cases that are equal to 4.41 and 2.62 $\mu$m, respectively. Moreover it appears that the extreme values indicated by the dashed vertical segments largely maximize the uncertainty of the retrieval. Except for the f22 event mean values of $Dw$ are larger than 2.37 $\mu$m. This is consistent with statistics on no-fog events whereby particle diameters as measured by WELAS do not exceed 2 $\mu$m.

As expected critical supersaturation associated are very low : 0.043 % for the median and 0.035-0.051 % for the 25th-75th percentiles, with only one case that exceeds 0.1 %. These values are close to the ones measured and modelled by Hudson (1980), Svenningsson et al. (1992), Ming and Russell (2004) and Hammer et al. (2014). Fig 8 shows the scatterplot of the hygroscopic parameter as a function of the critical supersaturation for each of the 23 fog events with their uncertainty interval. Though the scatter is rather large, the general trend points towards a slight decrease of $\kappa$ values as critical supersaturation values increase, which corresponds to the expected behaviour as depicted for example in Fig. 6 from CCN data. Since 80 % of the data fall between 0.13 and 0.27, it appears that the CCN ability of the aerosols sampled at SIRTA are very similar and could be considered as constant as suggested by Dusek et al. (2006). We therefore recommended to use $\kappa = 0.17 \pm 0.05$ for modelling studies on this area.

Finally the corresponding concentration of activated particles $N_{act}$ are 53.5 $cm^{-3}$ for the median and 28.5-111 $cm^{-3}$ for the 25th-75th percentiles. These values are slightly lower than the $N_{FM}$ ones derived from FM-100 data (reported in Fig. 3) which give a median and 25th-75th percentiles of 61 and 34-103 $cm^{-3}$, respectively. Indeed measurements at ambient humidity indicate that many supermicron particles are available. For instance the median of the average concentration values of particles with diameter in the range of [0.96-50] $\mu$m at the beginning of the fog events reaches 389 $cm^{-3}$ with 25th-75th percentiles of 260-660 $cm^{-3}$. It follows that only a small fraction of the fog hydrated particle correspond to activated droplets due to low critical supersaturations encountered. Hence the use of FM-100 data over the range [2-50] $\mu$m tends to overestimate the droplet concentration by taking into account non-activated aerosol particles. A noticeable exception comes from the f22 case with a maximum value of 264 $cm^{-3}$. Surprisingly, the median $N_{FM}$ value for this case is only 36 $cm^{-3}$ which is rather low. This is explained by the fact that this case also corresponds to the lowest critical diameter $Dw$=1.44 $\mu$m, therefore the contribution





from WELAS data is the most important and emphasizes the underestimation of the FM-100 in the first classes (see Fig. 2). Thus for cases with low $Dw$ WELAS data must be taken into account to avoid an underestimation of the concentration.

Obviously because of the sharp increase of the particle size distribution below the wet activation diameter, the uncertainty interval of $N_{act}$ is very broad. For instance this interval is twice the $N_{act}$ value itself on average but could be as high as a factor 6 for the f1 event. But as mentioned before, $N_{act_i}$ and $N_{act_f}$ are the extreme possible values derived by cumulating uncertainties on $\kappa$ and on the variability of measurements during the one hour time period.

## 4.2 Impact of aerosol particles on fog droplets concentration

Mean values of $N_{act}$ are reported in Fig. 9 as a function of the other activation parameters $SS_{peak}$, $D_d$ and $\kappa$ for the 23 fog events. As expected Fig. 9-a) shows that $N_{act}$ values increase with $SS_{peak}$. Note that even if there are quite large uncertainty intervals, they follow the same trend as the mean values. Compared to Fig. 3, they are not obvious differences between STL and RAD cases. As a result, the median values of $N_{act}$ are 40 and 58 $cm^{-3}$, and the 25th-75th percentiles are 25-115 and 33-109 $cm^{-3}$, for STL and RAD cases, respectively. Radiative cases have thus higher values of activated particles but the percentile intervals are rather similar and the difference is less pronounced than the factor of 2 obtained from $N_{FM}$ derived from the FM-100 measurements only. It follows that the estimation of the critical diameter and the use of WELAS measurements below this value is of crucial importance to derive accurate values of fog droplet number concentration.

In contrast to Fig. 9-b) that exhibits a clear decrease of $D_d$ as $N_{act}$ increases, Fig. 9-c) reveals no trend at all from the several values of $\kappa$. This means that size matters more than hygroscopicity for a particle to be activated. Indeed largest aerosol particles are activated at low SS independently of their $\kappa$ values. There is some scatter on Fig. 9-b) for values of $D_d \leq 0.4\mu$m which suggest that some variability of the dry aerosol size number distribution between the different cases occurs mainly below this threshold.

To remove the influence of the aerosol number concentration we then normalize $N_{act}$ by the number concentration of activable aerosols $N^*$, defined as the number concentration of aerosol particles with diameter $\geq$ 200 nm which corresponds to the smallest dry diameter of Table 3. Indeed $N^*$ is roughly proportional to the total number of particles but one order of magnitude lower. Figure 10 shows this ratio of activated particles over activable ones, a kind of CCN ratio, as a function of the supersaturation on the left and of the dry diameter on the right. A more pronounced relationship appears which emphasizes the strongest influence of the particle size compared to the chemistry on the ability of particles to act as CCN. These results are consistent with Fitzgerald (1973) and Andreae and Rosenfeld (2008) who concluded that size mattered more than chemistry for aerosol particles activation. Svenningsson et al. (1992) found that the chemistry is as important as size by pointing out the control of water uptake by the parts of soluble material. However soluble mass changes with the third power of particle diameter but only linearly with soluble fraction (Andreae and Rosenfeld, 2008). Different colors are used depending on the $\kappa$ values as indicated by the caption. Fig. 10-a) shows that, at a given supersaturation, the highest values of the CCN ratio are



associated with the highest values of the hygroscopic paramter, and that at a given CCN ratio, $\kappa$ decreases as the supersaturation
increases. But more samples are needed before a robust interpretation of that feature to be established.

Figure 11 in contrast reveals that almost no relationship exists between $N_{act}$ and $N^*$. This result demonstrates that the
concentration of fog droplets is roughly independent of the aerosol number concentration in opposite to the general trend
depicted by Fig. 3. Indeed supersaturations encountered in such fog are so low that for continental aerosols numerous CCN are
always available to nucleate and form fog droplets. This is also consistent with the very large dry diameter derived.

It follows that the number concentration of fog droplets is mainly controlled by the peak supersaturation with hardly any
influence from the aerosol background. This is illustrated in Fig. 12 where $N_{act}$ are plotted as a function of $SS_{peak}$ in log scale
(black diamond, same data as Fig. 9-a)) superimposed to the statistics of CCN measurements (grey diamond). Compared to
the compilation of CCN spectra reported in Andreae and Rosenfeld (2008) (their Fig. 2), our CCN data with typical values of
CCN ranging from 250 to 1000 $cm^{-3}$ at SS=0.1 % and from 1000 to 5000 $cm^{-3}$ at SS=0.5 % are spread between SCMS data
(Hudson and Yum, 2001) and continental cases. However we observe a strong decrease of $N_{act}$ for $SS_{peak} \leq 0.1\%$ similar to
the ASTEX data collected in maritime stratocumuli and to Hudson (1980) fog activation spectra measurements. This indicates
that only a very small fraction of CCN are activated for such low values of supersaturation.

These results has been fitted with three differents formulae commonly used in modelling. The classical CCN parameter-
isation proposed by Twomey (1959) : $N_{CCN}$=C$S^k$, where C represents the CCN number concentration at SS=1 % and the
parameter $k$ that varies significantly (Martins et al., 2009). The formula suggested by Ji et al. (1998) : $N_{CCN}$=N (1-exp(-B$S^k$))
where N is the total number concentration of CCN and B and k are empirical coefficients to be determined. And the more gen-
eral description of the activation spectra proposed by Cohard et al. (1998) : $N_{CCN}$=C$S_{v,w}^k$F$(\mu,\frac{k}{2},\frac{k}{2}+1;-\beta S_{v,w}^2)$ where C is
proportional to $N_{CCN}$ that would be activated when supersaturation tends to infinity and parameters k, $\mu$ and $\beta$ are adjustable
parameters depending on the aerosol properties. Figure 12 shows that the parameterization of Ji et al. (1998) (green line) with
parameters as indicated by the legend, better reproduces the decrease of $N_{act}$ for SS < 0.1 % compared to the Twomey expres-
sion (blue line) but the drop is not as sharp as in the data. In contrast the parameterization of Cohard et al. (1998) (red line)
provides the best fit of the data for lower values of SS.

Figure 13 shows the CCN concentration that would be activated at SS=0.1 % as a function of the aerosol activable concen-
tration for the 23 fog events. The expected strong correlation between activated particles and aerosol background then clearly
appears. This confirms that the apparent independence observed in Fig. 11 is simply due to the low supersaturation values
experienced in fogs. The symbol color depends on $\kappa$ as preceding figures. Data are aligned according to their hygroscopic pa-
rameter. For a given activable concentration the number of activated particles at SS=0.1 % increases with $\kappa$ which is consistent
with Fig. 10-a).

One can note in Fig. 11 that the highest values of $N_{act}$ decrease as $N^*$ increases which is a slighty surprising since we
expected that $N_{act}$ and $N^*$ to follow the same pattern. Indeed this feature is explained by the scatter plot of $SS_{peak}$ as a
function on $N^*$ as reported in Fig. 14 which reveals that as aerosol activable concentration increases highest $SS_{peak}$ tend





towards lower values and thus the range of possible $SS_{peak}$ becomes narrower. This suggests that the supersaturation reached in fog could be limited by the activable aerosol concentration : if numerous aerosol particles are available they are efficient enough to uptake the water vapor excess and therefore limit the supersaturation. Hudson (1980) found that aerosol particle concentration does not have a great effect on fog supersaturation. Our result however supports the conclusions drawn by the numerical study performed by Bott (1991) that the higher the particle concentration, the lower the supersaturation is. Note that

no trend appears with $\kappa$ in Fig. 11, which means that the sensitivity of this process to the hygroscopicity is very weak. The symbol size of the data points is proportional to the median diameter of the activable aerosol but bears not relation to it.

### 4.3   Impact of CCN concentration on fog microstructure

The mean fog droplet concentration $Nd\_1h$ averaged over the one hour time interval at the beginning of the fog event is plotted in Fig. 15-a) as a function of $Nact$. Obviously they are almost identical apart from small deviations that come from averaging

the $Nact$ values derived for both $\kappa$ values. As previously symbols depend on the fog type which doesn't reveal differences between radiative (red diamond) and stratus lowering (blue diamond) fogs during the formation phase.

Once droplets have been activated, they compete for the available water vapour: when their number increases the size they can reach by water vapour diffusion growth decreases. This inverse relationship between the number and the size of droplets is clearly depicted in Fig. 15-b) that displays the concentration $Nd\_1h$ vs. the corresponding mean diameter $Dm$. Indeed the

highest $Nd\_1h$ of 255 $cm^{-3}$ corresponds to the lowest Dm of 4 $\mu m$, and for fog events with $Nd\_1h > 120$ $cm^{-3}$ the mean diameter can not exceed 7 $\mu m$ while it reaches twice this value for concentration as low as 25 $cm^{-3}$. However some scatter appears for samples with $Nd\_1h < 50$ $cm^{-3}$. Symbol colour corresponds to the mean LWC values as indicated by the legend. Lowest values of $Dm$ are associated with lowest values of LWC which suggests that a lack of available liquid water could also limit the droplet grow, even when there are few of them.

During a fog life cycle several processes contribute to the droplet size distribution. Droplets can grow by water vapour diffusion, or by collision-coalescence with other droplets due to gravitational and turbulent motions. Conversely, they can evaporate if the supersaturation decreases due to heating of the air mass for example or in case of mixing with clear air. Figures 16-a) and b) show the same plots except that $Nd$, $Dm$ and $LWC$ correspond to mean values over the complete fog life cycle. The mean droplet concentration values are significantly lower especially for cases with high $Nact$ values. For instance for

cases with Nact > 50 $cm^{-3}$ the ratio $Nd\_cycle/Nact$ ranges from 0.25 to 0.92 with an average value of 0.58. Values reported in Brenguier et al. (2011) exhibit a clear separation between cumulus and stratocumulus clouds, with values from 0.32 to 0.56 (0.46 on average) and from 0.72 to 0.96 (0.87 on average), respectively, that were attributed to differences in entrainment-mixing processes in both cloud types. Here it appears that values are spanned over a large range with an intermediate average value. Worthy to be noted Fig. 16-b) reveals that $Nd\_cycle$ for a radiative fog can not exceed 70 $cm^{-3}$ which suggests that this

reduction is more pronounced in case of a radiative fog while STL cases seem less affected. Indeed, for cases with Nact > 100 $cm^{-3}$ the average $Nd\_cycle/Nact$ is 0.80 for a STL fog, a value close to stratocumulus clouds, while it drops down to 0.45 for RAD cases. This suggests that the processes involved in the reduction of the CDNC in a radiative fog are as efficient as the entrainment-mixing occurring in cumulus clouds. Note that many studies (Pilié et al., 1975; Choularton et al., 1981; Gerber,



1991; Bergot, 2013) among others, pointed out the key mechanism of the turbulent mixing in fogs. Figure 16-b) shows that mean diameter values averaged over the fog life cycle are similar to the previous ones. The resulting cluster of data points does not reveal any general trend and the anti-correlation between size and number of droplets is no longer noticeable. Indeed, the evolution of microphysical properties during the fog life cycle is complex and highly varies from case to case and depends on

many parameters. A comprehensive study on this topic has been performed and will appear in a forthcoming paper.

## 5 Discussion

In this experimental study we derived accurate estimations of the activated droplet concentration at the beginning of a fog event. This was done through a careful estimation of the critical supersaturation and of the wet critical diameter which allowed us to integrate the composite size distribution derived from WELAS and FM-100 measurements. We have shown that the derived

parameters are consistent with the parameterization of Cohard et al. (1998). They are also consistent with measurements reported in (Hudson, 1980) who developed the Isothermal Haze Chamber to extend the CCN measurements to supersaturation range below 0.1 % and derived fog condensation nuclei (FCN). Indeed he found effective supersaturation between 0.06 and 0.1 % as well as sharp changes in FCN spectra below 0.1 % with activated concentration at SS=0.4 % ranging from 0.8 $cm^{-3}$ over the sea to 250 $cm^{-3}$ for polluted cases.

To our knowledge, no other study has retrieved experimentally concentration of activated particles in fog. Usually measurements of droplet spectra reported in the literature were performed by single particle counters such as FSSP, CDP or FM-100 over the range [2-50] $\mu$m. The corresponding droplet concentrations are in the range of a tenth to several hundred $cm^{-3}$ (Wendisch et al., 1998; García-García et al., 2002; Gultepe and Milbrandt, 2007a; Niu et al., 2012; Price, 2011; Liu et al., 2011; Burnet et al., 2012; Lu et al., 2013; Zhao et al., 2013). In our cases, concentration values are rather low for semi-urban

conditions with only one event exceeding 150 $cm^{-3}$. For instance, Liu et al. (2011) and Niu et al. (2012) reported values recorded in Pancheng (China) as high as 800 and 1000 $cm^{-3}$, respectively. However we have shown that our resulting values of critical diameter are large with a median value of 3.8 $\mu$m. Consequently, the use of FM-100 data over the range [2-50] $\mu$m will lead to an overestimation of the droplet concentration, because is includes deliquescent aerosol particles that are not activated.

We have further shown that the droplet number concentration does not increase as the aerosol loading increases because the actual supersaturations reached in these types of fog are too low. Moreover we rather observe a decrease of droplet concentrations for the higher values of aerosol concentration which could be explained by a limitation of the supersaturation according to Bott (1991). In contrast some recent numerical simulations exhibit a strong positive correlation between aerosol and droplet concentrations (Zhang et al., 2014; Stolaki et al., 2015). For instance the sensitivity study of Stolaki et al. (2015) indicates that

the fog droplet concentration during the mature stage is 2.6 times greater (2.9 times lower) when doubling (halving) the CCN number concentration of the accumulation mode. Indeed at the fog onset, the droplet concentration at the surface reaches 467 $cm^{-3}$ in their simulation with a double CCN concentration. This value is much larger than that of our estimations which is $\leq$ 150 $cm^{-3}$ except in one case where it reaches 264 $cm^{-3}$. For their reference run also this value is as high as to 304 $cm^{-3}$. The





CCN activation scheme used by the model follows the expression of Cohard et al. (1998). Given the CCN activation spectrum derived from their values of CCN size distribution parameters (geometric mean radius and standard deviation of 0.1525 $\mu m$ and 2.33, respectively) and concentration (540 $cm^{-3}$) and solubility of aerosols (0.4), this corresponds to an estimation of the maximum supersaturation of 0.15 %. Even for activated concentration of 250 $cm^{-3}$ the corresponding SS is still $\simeq$ 0.1 %. We have shown that CCN measurement at SS=0.1 % exhibits a linear increase of the droplet concentration with the activable aerosol concentration. Consequently with such high supersaturation values it is consistent that their simulations produce a strong positive correlation between aerosol and droplet concentration.

More generally, one shall keep in mind that the use of the CCN activation spectra provides a satisfactory estimation of the activated concentration, providing that the maximum supersaturation is correctly diagnosed. For instance, a current limitation of schemes with adjustment to saturation with parameterized peak supersaturation is that the formulae do not take into account pre-existing liquid water in the model grid box (Thouron et al., 2012). This leads to a significant overestimation of the supersaturation peak value that in turn will overestimate the activated concentration. Moreover, in sensitivity study in which the CCN concentration is increased (or decreased) while keeping other parameters constant, intrinsically introduces a strong dependency between droplet and aerosol particles. Indeed, the pre-existing hygroscopic aerosol particles contribute to the sink term of water vapour during the radiative cooling that will also reduce the supersaturation peak value at the fog onset and then limits the number of activated particles. Thus changing the aerosol properties in models using such a scheme, provides a useful way to modify the droplet concentration and study the impact of this latter on the fog life cycle but it shall not be used to asses the impact of aerosol properties themselves on the fog life cycle.

## 6 Conclusions

In situ microphysical measurements collected during 23 wintertime fog events sampled from October 2010 to March 2013 at the semi-urban site SIRTA near Paris have been examined to document their properties. They reveal a large variability of their characteristic values in terms of number concentration and size of fog droplets depending on the different cases, as well as various evolution of these properties during the fog life cycle. The aerosol background exhibits also a highly variable number concentration of particles > 10 $nm$ before the fog onset. The objective of this paper was to evaluate the impact of aerosol particles on the fog microphysics. As a first step, we focused on the relationship between aerosol and fog droplet number concentrations as we expected that they follow the same trend with more numerous fog droplets with an increase of the aerosol loading.

To derive accurate estimations of the actual activated fog droplet number concentration ($Nact$), we determined the hygroscopicity parameter, the dry and wet critical diameters, and the critical supersaturation for the 23 events by using an iterative procedure based on the $\kappa$-Köhler theory which combined CCN measurements, dry particle distribution from an SMPS and composite wet particle size distribution at ambient humidity derived from WELAS and FM-100 measurements. These data are averaged over a one hour time period before and during the fog onset to characterize the air mass and the fog properties, respectively.




Values of the hygroscopicity parameter $\kappa$ extrapolated at SS$\simeq 0.05\%$ were found to range from 0.09 to 0.3 which are characteristic for urban aerosol. They are rather similar from case to case thus we recommended to use $\kappa = 0.17 \pm 0.05$ for fog modelling studies in this urban area. Our study reveals low values of the derived critical supersaturations with median of 0.043 % and 25th-75th percentiles of 0.035-0.051 %. Consequently wet and dry activation diameters are high and the corresponding $Nact$ are low for continental conditions with a median concentration of 53.5 $cm^{-3}$ and 25th-75th percentiles of 28.5-111 $cm^{-3}$. Our results depict a sharp drop of $Nact$ as critical supersaturations decrease which is best fitted by the parameterization of Cohard et al. (1998).

No detectable trend between concentration of aerosol particles with diameter > 200 $nm$ and $Nact$ was observed. In contrast the CCN data at 0.1 % supersaturation exhibits a strong correlation with these aerosol concentrations. We therefore conclude that the droplet number concentration is roughly independent of the aerosol one because the actual supersaturations reached in these fog events are too low. Unfortunately, no measurement of dry aerosol particles > 500 nm were available to investigate the contribution of the largest particles. Our analysis corroborates modelling study by Bott (1991) suggesting that an increase of the aerosol concentration limits the SS values. Moreover it is found that the activated fraction mainly depends on the aerosol size while the chemistry, as represented by the hygroscopic parameter, appears to be of a secondary importance.

Despite that a stratus lowering fog appears to be associated with lower aerosol loading than with a radiation fog event, no significant differences were observed with respect to the droplet number concentration at the fog formation when calculations are performed by taking into account the wet critical diameter. In contrast, statistics over the complete life cycle indicate that a radiation fog is subject to a pronounced decrease in the droplet concentration while stratus lowering cases seem unaffected. In a radiation fog this decrease, which occurs mainly for events with $Nact > 50\ cm^{-3}$, is significant with a ratio of average droplet concentration to $Nact$ ranging from 0.25 to 0.67. This reduction is similar to the dilution which results from entrainment-mixing in cumulus clouds. The expected inverse relationship between the number and the size of the droplets at the formation phase is clearly depicted except for some cases with low liquid water content values. However this trend is less pronounced on average over the fog life cycle.

*Acknowledgements.* Authors are very grateful to all SIRTA operators and database managers. This campaign was held in the framework of the PreViBOSS project supported by DGA/DGIS. This research was partially funded by the European Community's Seventh Framework Program (FP7/2007-2013) under the SESAR WP 11.2.2 project, under Grant Agreement 11-120809-C.



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




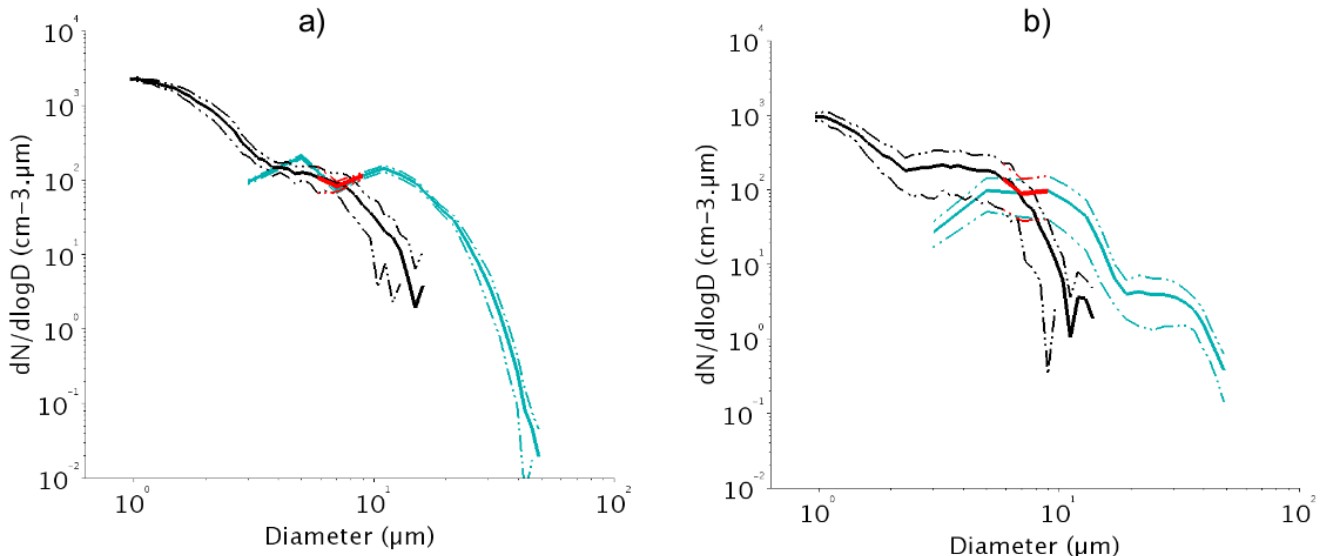

**Figure 1.** One hour average composite number size distributions at ambient humidity derived from WELAS (black) and FM-100 (cyan) at the beginning of the fog event for cases a) f6 and b) f20. The solid line represents mean value and the dotted lines mean ± one standard deviation. The red segments correspond to the result of merging both distributions between 6 and 8 $\mu$m.

**Table 1.** Instrumentation deployed during the campaigns that are used for this study.

| Instruments | Measured parameters | Time resolution |
|---|---|---|
| **P**ALAS WELAS - 2000 | Hydrated and activated aerosol particles number size distribution D=[0.39-42] $\mu$m | 5 min |
| **D**MT Fog Monitor | Droplet number size distribution D=[2-50] $\mu$m | 1 min |
| **P**VM Gerber | Liquid water content D=[3-50] $\mu$m | 1 min |
| **D**egreanne DF20+ | Horizontal visibility | 1 min |
| **T**SI SMPS | Dry aerosol particles distribution D=[10.6-493] nm | 5 min |
| **C**CNC DMT | CCN concentration SS=[0.1-0.5] % | 20 min for a cycle |





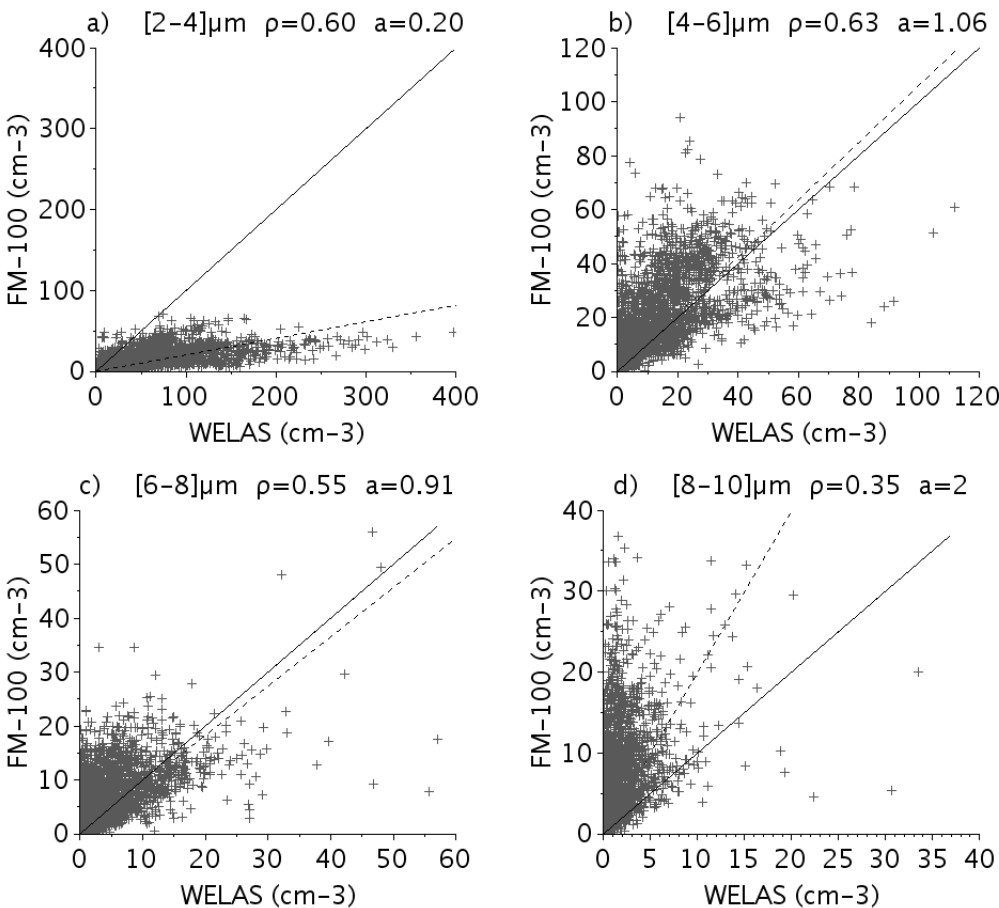

**Figure 2.** Scatterplot of the five minutes average particle number concentration values as measured by the FM-100 for the four first bins vs the integrated WELAS measurement over the corresponding diameter range a) [2-4] $\mu$m, b) [4-6] $\mu$m,c) [6-8] $\mu$m and d) [8-10] $\mu$m. Solid line corresponds to 1:1 line and dashed line corresponds to best fit line with correlation coefficient $\rho$ and slope $a$, as indicated in the legend.





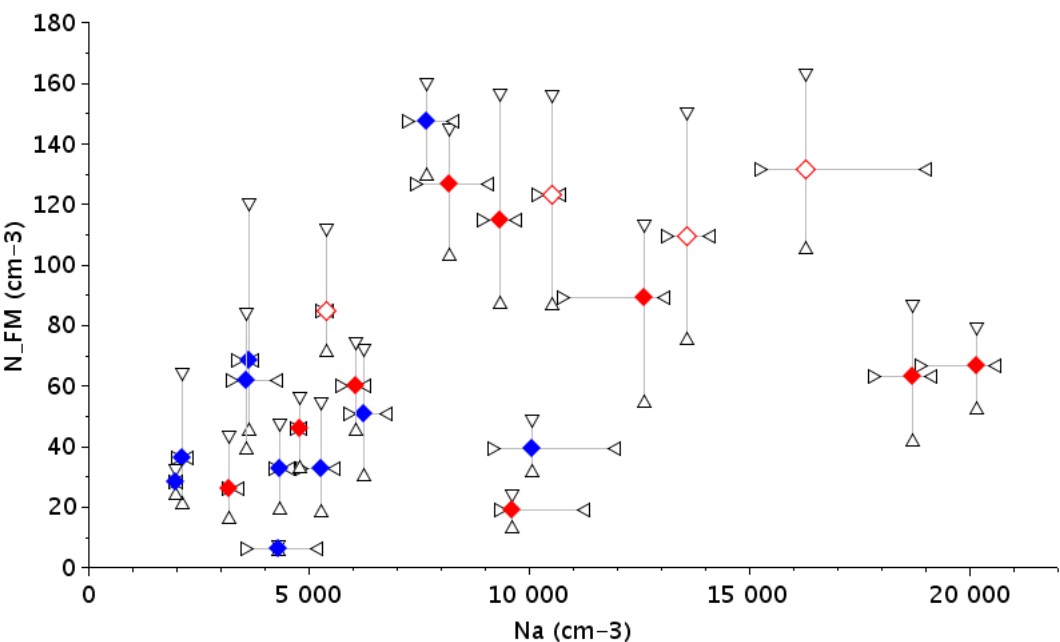

**Figure 3.** Droplet number concentration as derived from the FM-100 collected during the whole fog event as a function of the aerosol number concentration as derived from the SMPS measurements collected over the last hour before the fog beginning. Median values (diamond) are indicated for each case of table 2 with the 25th and 75th percentiles (triangle) to represent the variability. Symbols depend on the fog type STL (blue) and RAD (red) with open and solid symbols for thin and developed fogs, respectively.

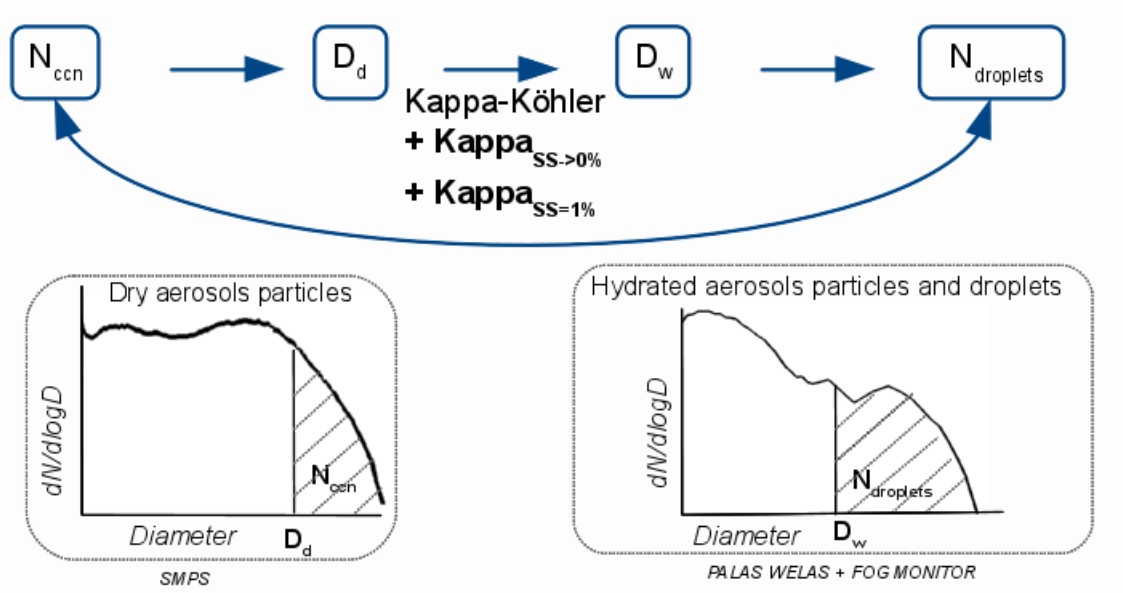

**Figure 4.** Schematic of the iterative method used to retrieve fog activation properties from dry and ambient humidity particle size distributions.

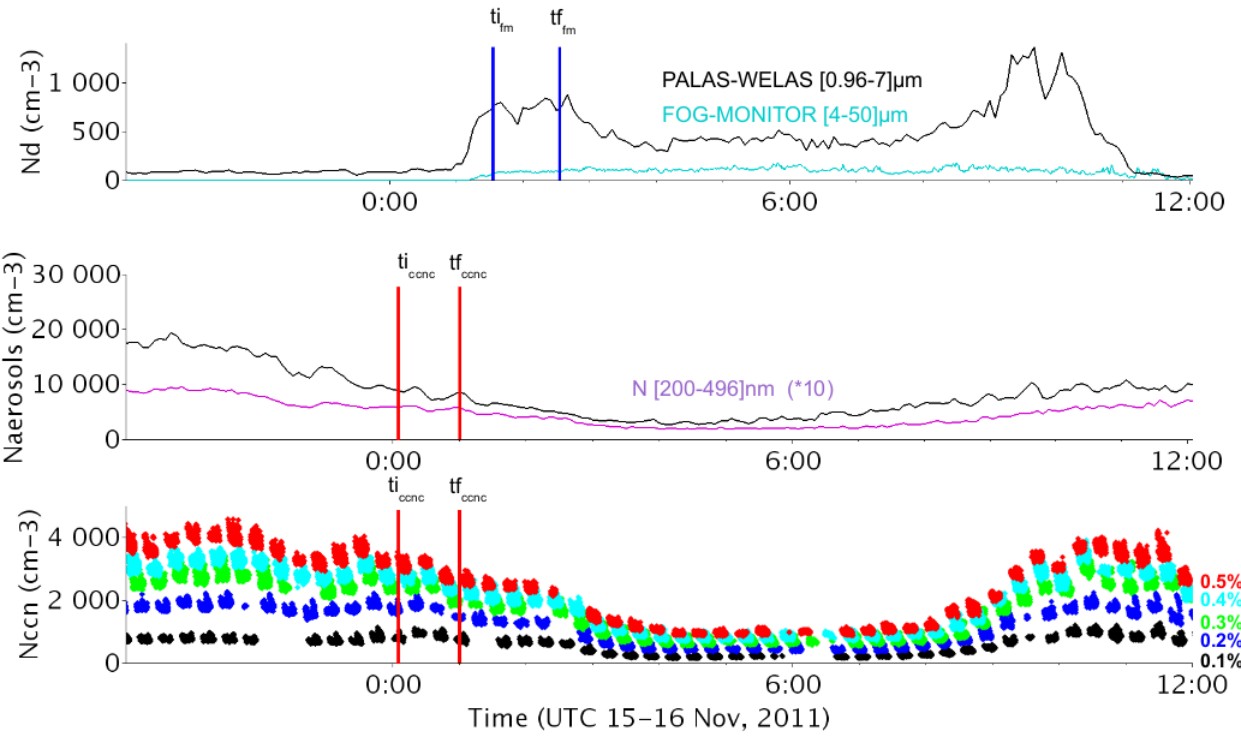

**Figure 5.** Time series of measurements for the f6 case : (a) particle number concentration at ambient humidity from FM-100 (cyan) and WELAS (black); (b) dry particle number concentration from SMPS : total (black) and particle with diameter > 200 nm (purple); and (c) Nccn from the CCN chamber at each supersaturation with color as indicated on the label. Vertical segments indicate the selected time periods before (red) and during (blue) the beginning of the fog event.



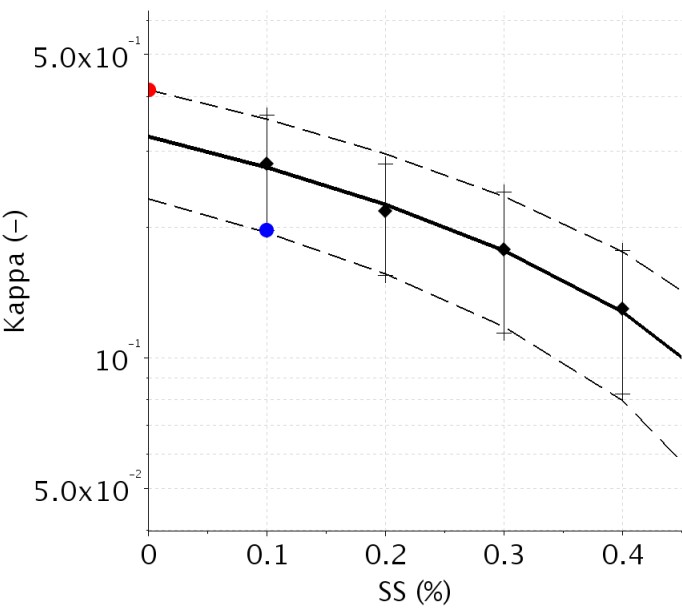

**Figure 6.** Derived $\kappa$ values as a function of CCN chamber supersaturations from measurements over the 1 hour time period before the fog event for the f6 case. Mean (diamond, solid line) and mean $\pm$ one standard deviation (dashed lines). Extrapolated $\kappa_{inf}$ (blue dot) and $\kappa_{sup}$ (red dot) are also indicated.



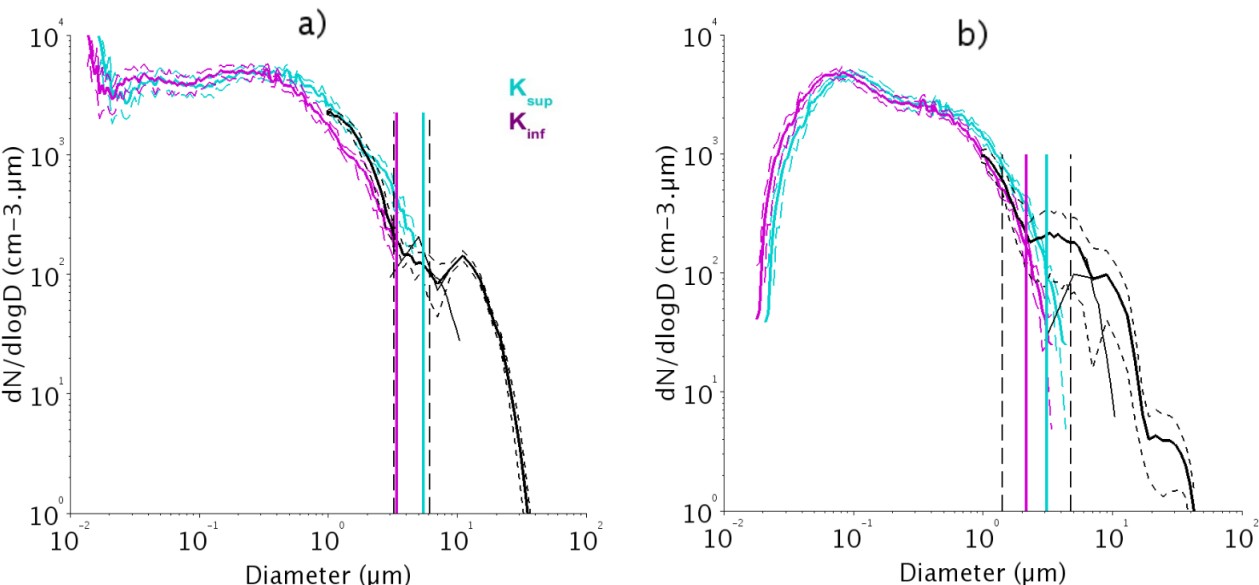

**Figure 7.** One hour wet number size distribution before the fog event resulting from the hygroscopic grow at RH=100 % of the dry distribution measured by the SMPS calculated with $\kappa_{inf}$ (purple) and $\kappa_{sup}$ (cyan) for case a) f6 and b) f20. The in-fog composite size distribution of Fig. 1 is superimposed (black). The solid lines represent mean values and the dashed lines mean ± one standard deviation. The vertical segments indicate the mean values of $D_w$ corresponding to each $\kappa$ (solid color lines) and the extreme values (dashed lines).

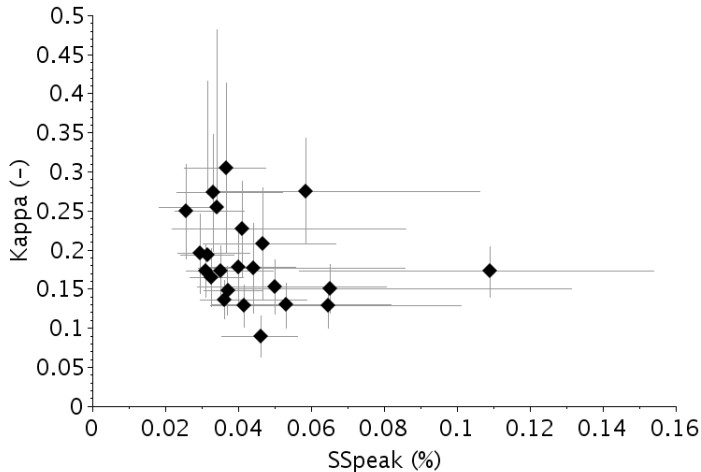

**Figure 8.** Hygroscopicity parameter $\kappa$ as a function of the critical supersaturation for the 23 fog events. Diamonds correspond to mean values and error bars to uncertainty intervals.





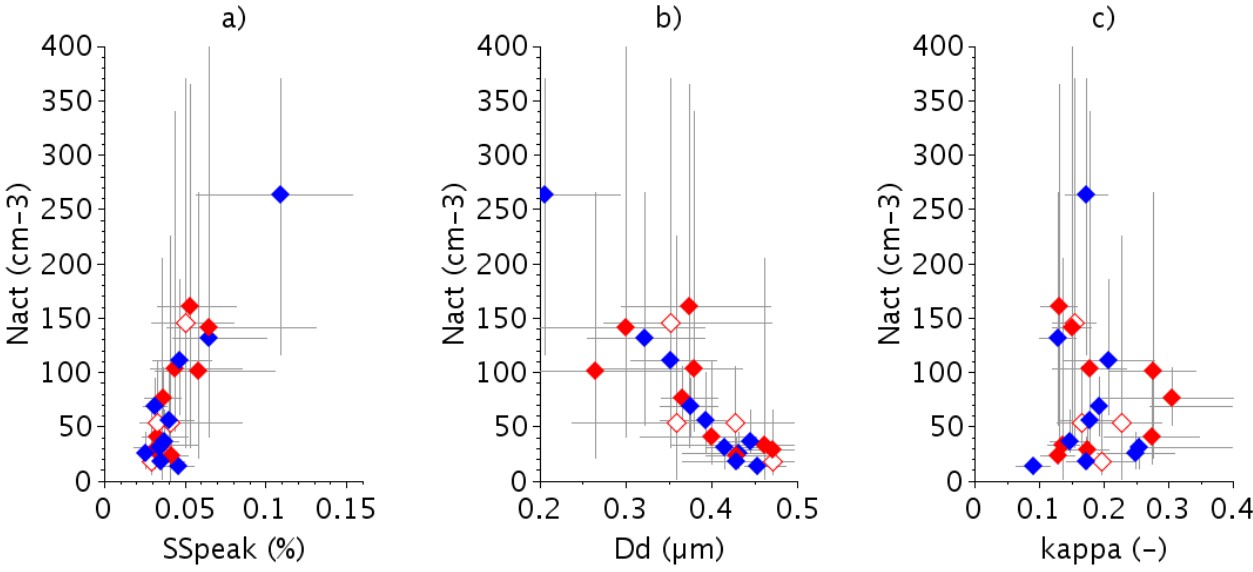

**Figure 9.** Number concentration of activated particles $N_{act}$ as a function of a) the critical supersaturation, b) the dry critical diameter and c) $\kappa$, for the 23 fog cases. Diamonds correspond to mean values and error bars to uncertainty intervals. Same symbols as in Fig. 3.

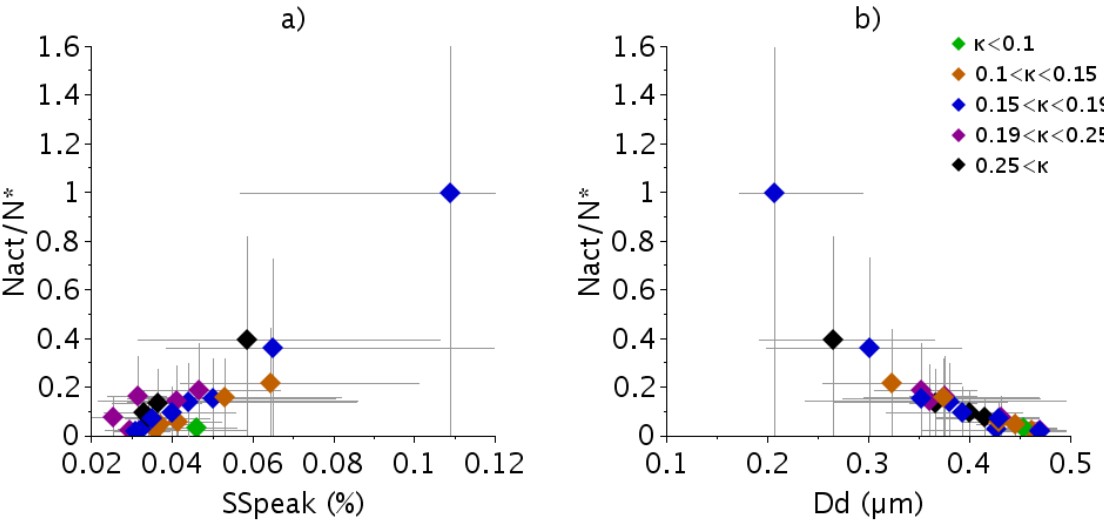

**Figure 10.** Ratio of activated particles to concentration of aerosol particles with diameter > 200 nm for the 23 fog cases function of a) the critical supersaturation and b) the dry critical diameter. Diamonds correspond to mean values and error bars to uncertainty intervals. Color as a function of $\kappa$ as indicated in the legend.





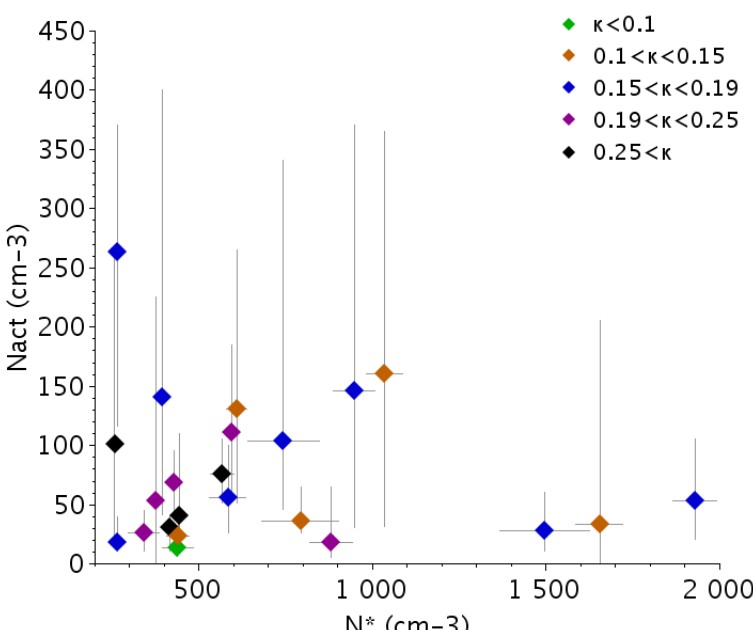

**Figure 11.** Number concentration of activated particles $N_{act}$ as a function of $N^*$ the concentration of aerosol particles with diameter > 200 nm for the 23 fog events. For $N^*$ diamonds correspond to median values and error bars to 25th-75th percentile intervals. Colour as a function of $\kappa$ as indicated in the legend.





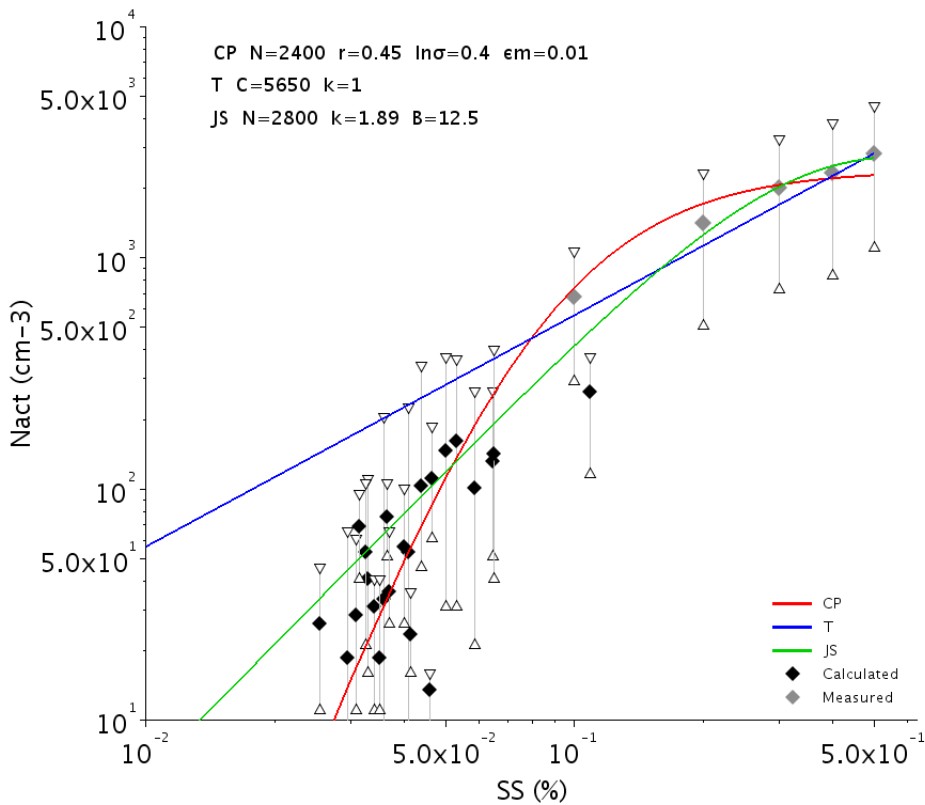

**Figure 12.** Number concentration of activated particles $N_{act}$ as a function of the critical supersaturation for the 23 fog events (black) superimposed for the statistics of CCN chamber measurements (grey). For CCN measurements diamonds correspond to median values and error bars to 25th-75th percentile intervals. Colour lines correspond to fitted results of $N_{CCN}=CS_{v,w}^k F(\mu, \frac{k}{2}, \frac{k}{2}+1; -\beta S_{v,w}^2)$ (red), $N_{CCN}=CS^k$ (blue) and $N_{CCN}=N$ (1-exp(-B$S^k$)) (green), with parameters as indicated on the legend.





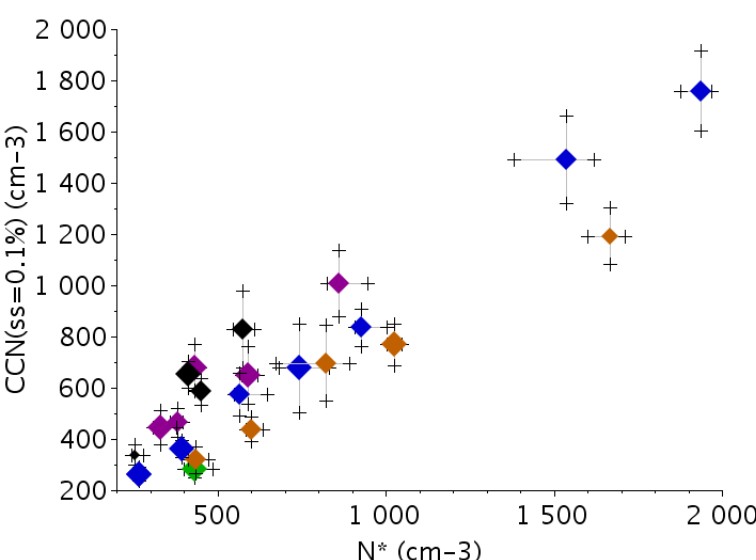

**Figure 13.** Median values for the 23 fog events of the number concentration measured by the CCN chamber at $SS$= 0.1 % as a function of the concentration of aerosol particles with diameter > 200 nm. Colour as a function of $\kappa$ as in Fig. 11. Symbol size proportional to the mean diameter.




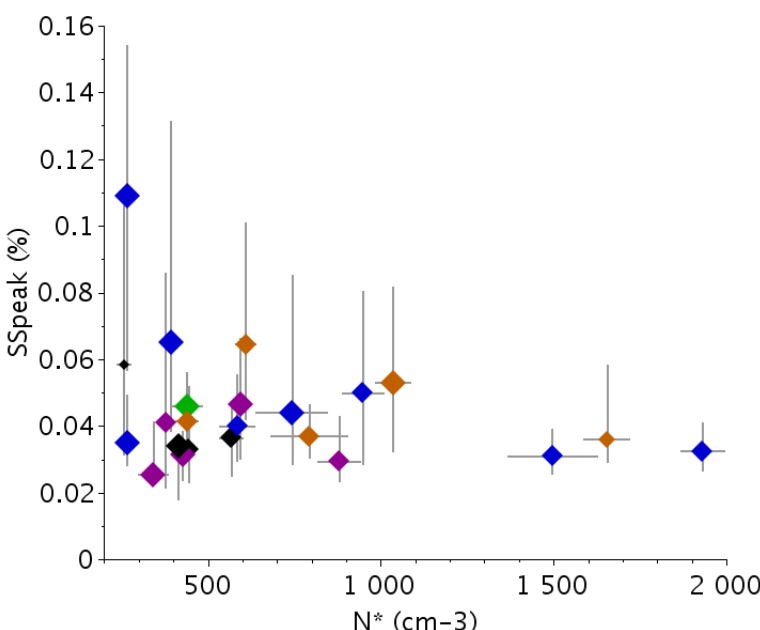

**Figure 14.** Critical supersaturation as a function of the concentration of aerosol particles with diameter > 200 nm for the 23 fog cases. Diamonds correspond to mean values and error bars to uncertainty intervals. Colour and size of symbols as in Fig. 13.





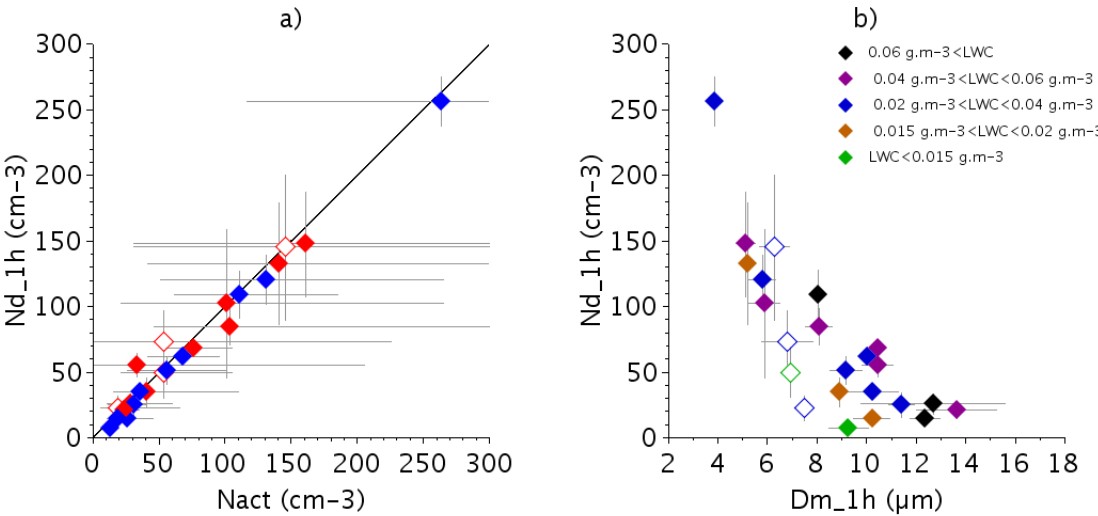

**Figure 15.** Droplet number concentration at the beginning of the fog for the 23 cases as a function of a) the activated particle concentration $N_{act}$ and b) the mean fog diameter. Diamonds correspond to median values and error bars to 25th-75th percentile intervals. Symbol colors as a function of a) fog type as in Fig. 3 and b) LWC values as indicated on the legend.

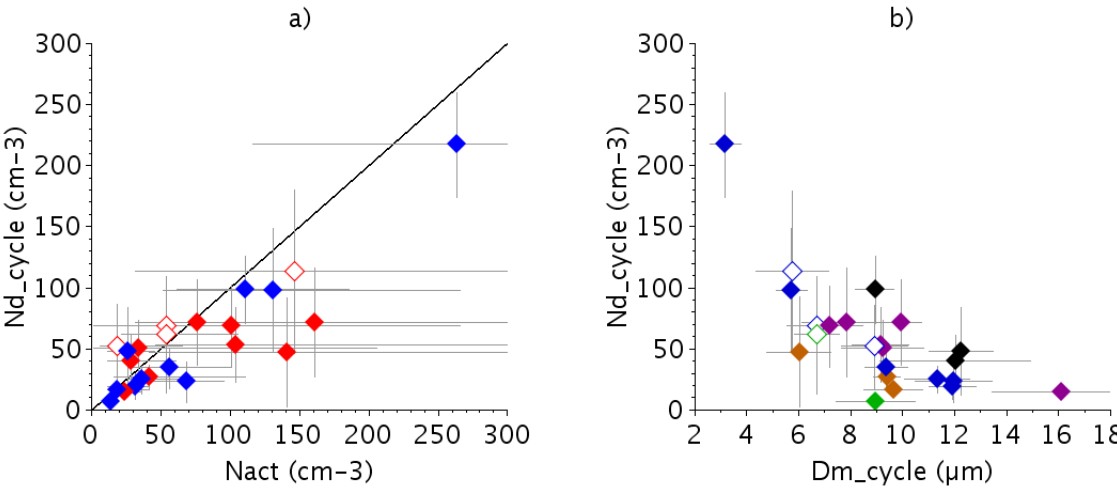

**Figure 16.** Same as Fig. 15 except that statistics of $Nd$, $Dm$ and LWC are computed over the complete fog life cycle.





**Table 2.** List of fog events analysed here. Type RAD corresponds to radiation fog and STL to stratus lowering, "thick" to fog developed on the vertical, and "thin" to fog layer with top altitude lower than 18 m. $N_a$ is the number concentration of aerosol particles derived from SMPS data. 25th, 50th and 75th percentiles are computed from the distribution of 5 minutes samples over the last hour before the fog beginning. $N_{FM}$ is the droplet number concentration as measured by the FM-100 over the range [2-50] $\mu$m. Statistics of $N_{FM}$ are computed with minute average data of samples with LWC > 0.005 $g.m^{-3}$ collected during the whole fog event.

| Fog ID | Start time [UTC] | End time [UTC] | Type (1) | Type (2) | $N_a$ ($cm^{-3}$) 25th | $N_a$ ($cm^{-3}$) 50th | $N_a$ ($cm^{-3}$) 75th | $N_{FM}$ ($cm^{-3}$) 25th | $N_{FM}$ ($cm^{-3}$) 50th | $N_{FM}$ ($cm^{-3}$) 75th |
|---|---|---|---|---|---|---|---|---|---|---|
| f1 | 16/11/10 2200 | 17/11/10 0540 | RAD | thick | 18868 | 20175 | 20625 | 52 | 67 | 80 |
| f2 | 19/11/10 0540 | 19/11/10 1010 | STL | thick | 5877 | 6235 | 6783 | 30 | 51 | 73 |
| f3 | 19/11/10 1540 | 19/11/10 1750 | STL | thick | 3540 | 4296 | 5213 | 5.3 | 6.3 | 7.4 |
| f4 | 10/11/11 1800 | 11/11/11 1730 | RAD | thick | 10723 | 12602 | 13086 | 5 4 | 89 | 114 |
| f5 | 15/11/11 0230 | 15/11/11 0940 | RAD | thick | 5703 | 6043 | 6285 | 45 | 60 | 75 |
| f6 | 16/11/11 0110 | 16/11/11 1330 | RAD | thick | 7391 | 8188 | 9087 | 103 | 127 | 145 |
| f7 | 16/11/11 1600 | 17/11/11 0010 | STL | thick | 7213 | 7648 | 8310 | 129 | 147 | 160 |
| f8 | 18/11/11 0130 | 18/11/11 0410 | RAD | thin | 5245 | 5396 | 5447 | 71 | 85 | 112 |
| f9 | 19/11/11 2200 | 20/11/11 0830 | RAD | thin | 13094 | 13587 | 14135 | 75 | 110 | 150 |
| f10 | 21/11/11 2350 | 22/11/11 0810 | RAD | thin | 10160 | 10526 | 10737 | 87 | 123 | 156 |
| f11 | 22/11/11 2050 | 22/11/11 2230 | RAD | thin | 15209 | 16290 | 19031 | 105 | 132 | 164 |
| f12 | 23/11/11 0325 | 23/11/11 1005 | RAD | thick | 8920 | 9320 | 9723 | 87 | 115 | 157 |
| f13 | 24/11/11 0620 | 24/11/11 1400 | STL | thick | 4770 | 5258 | 5586 | 18 | 33 | 55 |
| f14 | 24/11/11 1610 | 24/11/11 1815 | STL | thick | 4180 | 4340 | 4566 | 19 | 33 | 48 |
| f15 | 25/11/11 2140 | 26/11/11 1030 | STL | thick | 3328 | 3618 | 3762 | 45 | 68 | 121 |
| f16 | 28/11/11 0630 | 28/11/11 1040 | RAD | thick | 9308 | 9618 | 11258 | 13 | 19 | 24 |
| f17 | 16/11/12 2045 | 17/11/12 0920 | STL | thick | 9145 | 10051 | 11959 | 31 | 39 | 49 |
| f18 | 20/11/12 0300 | 20/11/12 0910 | RAD | thick | 3108 | 3168 | 3410 | 16 | 26 | 44 |
| f19 | 20/11/12 2015 | 20/11/12 2250 | STL | thick | 3198 | 3567 | 4282 | 39 | 62 | 84 |
| f20 | 22/11/12 0315 | 22/11/12 0910 | RAD | thick | 4673 | 4780 | 4827 | 33 | 46 | 56 |
| f21 | 30/11/12 1900 | 01/12/12 0245 | RAD | thick | 17791 | 18712 | 19151 | 42 | 63 | 87 |
| f22 | 10/01/13 0215 | 10/01/13 0336 | STL | thick | 1975 | 2098 | 2273 | 21 | 36 | 64 |
| f23 | 10/01/13 0500 | 10/01/13 0640 | STL | thick | 1904 | 1955 | 2009 | 24 | 29 | 33 |





**Table 3.** Activation properties of the 23 fog events : hygroscopicity parameters $\kappa_{inf}$ and $\kappa_{sup}$, mean values and uncertainty intervals of the dry diameter $D_d$, the wet diameter $D_w$, the supersaturation $SS_{peak}$ and the number concentration of activated particles $N_{act}$. 5th, 25th, 50th, 75th and 95th percentiles of the distribution of the 23 cases are indicated on the last lines for each parameter.

| Fog ID | $\kappa_{inf}$ | $\kappa_{sup}$ | $D_d$ μm | $D_{di}$ μm | $D_{ds}$ μm | $D_w$ μm | $D_{wi}$ μm | $D_{ws}$ μm | $SS$ % | $SS_i$ % | $SS_s$ % | $N_{act}$ cm$^{-3}$ | $N_{acti}$ cm$^{-3}$ | $N_{acts}$ cm$^{-3}$ |
|---|---|---|---|---|---|---|---|---|---|---|---|---|---|---|
| f1 | 0.11 | 0.16 | 0.46 | 0.35 | 0.50 | 4.24 | 2.56 | 5.14 | 0.04 | 0.03 | 0.06 | 33.5 | 1 | 206 |
| f2 | 0.10 | 0.16 | 0.32 | 0.25 | 0.39 | 2.42 | 1.49 | 3.57 | 0.06 | 0.04 | 0.10 | 131 | 51 | 266 |
| f3 | 0.06 | 0.12 | 0.45 | 0.44 | 0.49 | 3.31 | 2.66 | 4.25 | 0.05 | 0.04 | 0.06 | 13.5 | 6 | 16 |
| f4 | 0.12 | 0.24 | 0.38 | 0.26 | 0.44 | 3.65 | 1.73 | 5.15 | 0.04 | 0.03 | 0.09 | 104 | 46 | 341 |
| f5 | 0.20 | 0.35 | 0.4 | 0.32 | 0.45 | 4.83 | 2.89 | 6.56 | 0.03 | 0.02 | 0.05 | 41 | 16 | 111 |
| f6 | 0.20 | 0.41 | 0.37 | 0.34 | 0.41 | 4.41 | 3.19 | 6.08 | 0.04 | 0.02 | 0.05 | 76 | 51 | 106 |
| f7 | 0.14 | 0.28 | 0.35 | 0.31 | 0.41 | 3.46 | 2.27 | 5 | 0.05 | 0.03 | 0.07 | 111 | 61 | 186 |
| f8 | 0.17 | 0.29 | 0.36 | 0.24 | 0.50 | 3.79 | 1.73 | 6.9 | 0.04 | 0.02 | 0.09 | 53.5 | 1 | 226 |
| f9 | 0.14 | 0.25 | 0.47 | 0.39 | 0.50 | 5.22 | 3.43 | 6.37 | 0.03 | 0.02 | 0.04 | 18.5 | 6 | 66 |
| f10 | 0.12 | 0.19 | 0.35 | 0.27 | 0.47 | 3.03 | 1.83 | 5.15 | 0.05 | 0.03 | 0.08 | 146 | 31 | 371 |
| f11 | 0.13 | 0.20 | 0.43 | 0.42 | 0.49 | 4.67 | 3.59 | 5.63 | 0.03 | 0.03 | 0.04 | 53.5 | 21 | 106 |
| f12 | 0.14 | 0.21 | 0.47 | 0.42 | 0.50 | 4.9 | 3.76 | 5.84 | 0.03 | 0.03 | 0.04 | 28.5 | 11 | 61 |
| f13 | 0.17 | 0.22 | 0.34 | 0.24 | 0.38 | 3.25 | 1.73 | 4.04 | 0.05 | 0.04 | 0.09 | 106 | 71 | 281 |
| f14 | 0.13 | 0.38 | 0.41 | 0.34 | 0.47 | 4.91 | 2.66 | 7.25 | 0.03 | 0.02 | 0.06 | 31 | 11 | 81 |
| f15 | 0.19 | 0.31 | 0.43 | 0.37 | 0.47 | 5.15 | 3.54 | 6.6 | 0.03 | 0.02 | 0.04 | 21 | 11 | 46 |
| f16 | 0.10 | 0.16 | 0.43 | 0.39 | 0.47 | 3.69 | 2.87 | 4.65 | 0.04 | 0.03 | 0.05 | 23.5 | 16 | 36 |
| f17 | 0.12 | 0.18 | 0.45 | 0.41 | 0.47 | 4.15 | 3.23 | 4.97 | 0.04 | 0.03 | 0.05 | 36 | 26 | 66 |
| f18 | 0.12 | 0.18 | 0.30 | 0.20 | 0.39 | 2.37 | 1.13 | 3.86 | 0.07 | 0.04 | 0.13 | 141 | 41 | 401 |
| f19 | 0.13 | 0.22 | 0.39 | 0.34 | 0.44 | 3.82 | 2.66 | 5.02 | 0.04 | 0.03 | 0.06 | 56 | 26 | 101 |
| f20 | 0.21 | 0.34 | 0.26 | 0.19 | 0.37 | 2.62 | 1.4 | 4.75 | 0.06 | 0.03 | 0.11 | 101 | 21 | 266 |
| f21 | 0.10 | 0.16 | 0.37 | 0.29 | 0.47 | 3.03 | 1.85 | 4.67 | 0.05 | 0.03 | 0.08 | 161 | 31 | 366 |
| f22 | 0.14 | 0.21 | 0.21 | 0.17 | 0.29 | 1.44 | 0.98 | 2.65 | 0.11 | 0.06 | 0.15 | 264 | 116 | 371 |
| f23 | 0.14 | 0.21 | 0.43 | 0.37 | 0.47 | 4.27 | 3.03 | 5.33 | 0.04 | 0.03 | 0.05 | 18.5 | 11 | 41 |
| 5th | 0.14 | 0.21 | 0.43 | 0.37 | 0.47 | 4.27 | 3.03 | 5.33 | 0.04 | 0.03 | 0.05 | 18.5 | 11 | 41 |
| 25th | 0.12 | 0.18 | 0.35 | 0.25 | 0.41 | 3.03 | 1.73 | 4.65 | 0.03 | 0.02 | 0.05 | 28.5 | 11 | 66 |
| 50th | 0.13 | 0.21 | 0.39 | 0.34 | 0.47 | 3.79 | 2.66 | 5.14 | 0.04 | 0.03 | 0.06 | 53.5 | 21 | 111 |
| 75th | 0.17 | 0.29 | 0.43 | 0.39 | 0.49 | 4.67 | 3.19 | 6.08 | 0.05 | 0.03 | 0.09 | 111 | 46 | 281 |
| 95th | 0.21 | 0.41 | 0.47 | 0.44 | 0.50 | 5.22 | 3.76 | 7.25 | 0.11 | 0.06 | 0.15 | 264 | 116 | 401 |