# Peer review of "Experimental study of the aerosol impact on fog microphysics."

_Atmospheric Chemistry and Physics, 2016_

## Referee Comment (RC1) · Anonymous Referee #2 · 24 Jun 2016

Rev. A Manuscript title: Experimental study of the aerosol impact on fog microphysics Date: June 24 2016 Decision: rejection

General comments: Overall objective of this work is to evaluate aerosol effect on fog microphysics based on observations and develop accurate estimation of Nd as a function of hygroscopicity parameter, size, and Sw. They state that high Na limits Sw, and activation is mainly function of aerosol size, not hygroscopicity. There are issues with text flow and organization that affects the quality of the work. Also, I found out it was difficult to follow the text because of issues related to proper use of definitions (activation of droplets) and missing info on the figs captions. Other point, authors should clearly show how different their findings compared to others.

Although a decision as rejection is given, this paper, based on corrections suggested

below, can be considered for a possible publication.

Major issues: 1. This paper is not organized properly, flow in the text for scientific steps are not clear, worst part of this manuscript is use of terminology that includes "activated droplet", and CCN, Nd, CN etc. Droplets are already activated. I suggest that authors prepare an abbreviation list and use the terms in the paper function of these definitions. Not easy to follow up with sentences if definitions are not clear. Pg 15; line 29, but also in many places. 2. Incomplete sentences are exist. 3. Need to clearly explain sensor differences and how do you use them. 4. Results are given in method section, and method is not clear. 5. Based on 1, you need to define CN, CCN, Na and their size ranges. 6. Nucleation happens usually >0.1 micron and up to 1-2 micron. Nd versus Na should represent this but I don't see a relationship on this. 7. Integration of WELAS and FM100 is a problematic, as seen from the plots (Fig. 1 and 2). They cant be integrated because of large differences in spectral distribution. 8. Did you compare LWC from gerber and FM device? This will show accuracy of FMD. 9. FMD and FMW should be compared for entire size range. 10. Fig. 2 shows clearly that FMD or WELAS Nd are really bad, no obs fit well into 1/1 line. How can use this data in an integrated way and make judgements. Better use them differently. 11. Your findings are very similar to previous studies, how different your work compared to others and why? 12. Fig. 3; why you use SMPS for entire size range? Activated Na are usually >0.1 micron? 13. Figs, should state which sensor, sampling rate, and size intervals and be consistent; for example Ndfmd and Ndw, NaSMPS etc. 14. Fig. 5; Nccn from CCN chamber; need to define this, when I look at the plots, I cant understand the trends because of definition issues. 15. Fig. 8; how did you get SSpeak? Explain in title or in an abbreviation list. 16. Fig. 7; why 1-hr time period is used? 17. Fig. 8; SSpeak increases, I expect Kappa increases, because more vapor in the air, aerosols should get more wet, why doesn't show this. 18. Fig. 9; SS=0.03; Nd=0????? Nact means what? Nact=Nd? Nact=CCN, what???? 19. Fig. 9c; kappa increases and Nact (CCN, Nd??) increases, it is ok, then, why kappa doesn't change with SS? 20. Fig10; b) Dd increases but not activation ratio?, I thought larger the size activation can

[Figure]

be more, am I wrong? 21. Fig. 10a; activation ratio doesn't change with SS, or very weak; I expect an increase, is it data analysis problem. 22. Fig. 11 is not clear, and Nact (is it CCN, or Nd). Why there is no correlation between N* and Nact? Why you expect a relationship? 23. Fig. 12; whose equations are given there??? 24. It is nice plot, probably you should focus on this plot, and reduce others. Nact means what it is Nd? Or CCN, you need to clarify your text, and use either Nact or Nd???? 25. Fig. 13; y axis now you say CCN (size range? >0.2 micron) and versus N* (<0.5 micron); this means CCN and N* have same channels; therefore, you see a correlation, explain this in discussions or correct the plot. 26. Fig. 15; now seen that Nd=Nact; then move this plot up and explain it. 27. Fig. 15b; Nd versus D; why you use 1 hr time period for this plot; be consistent with data analysis criteria. 28. Fig. 16a; shows Nd<Nact? You need to explain diff between these parameters in the method section. 29. Finally, conclusions should be short, listed, and importance be given.

---

## Referee Comment (RC2) · K. Noone (Referee) · 27 Jun 2016

Review of the manuscript

Experimental study of the aerosol impact on fog microphysics

by M. Mazoyer et alia

Kevin J. Noone, Stockholm University

**Overall**

This paper presents interesting measurements of aerosol-cloud interactions in fogs in a suburban location. The results and data are useful in terms of better understanding the life cycle of fog in relatively polluted locations. I feel that there are a couple of issues that need further explanation and discussion before the paper is ready for publication; I pointed these issues out in my first review, but they remain unaddressed in this version of the manuscript. I will repeat and expand on them here. The grammar has been improved.

**Major comments**

1)      Page 4, 1st paragraph

One of my main concerns with the paper from the pre-discussion version remains unaddressed in this version as well.

There is typically a rather substantial discrepancy between the concentrations of fog droplets measured by the WELAS and FM-100 instruments in the size range in which they overlap. The authors choose to believe the WELAS instrument up to 6 µm, the FM-100 above 8 µm, and average the two in the region between 6-8 µm. They cite Elias et al. (2015) as a reference for this choice, and do not provide further explanation. Unfortunately, the Elias et al. (2015) reference doesn't provide much justification for this choice, only citing another reference (Elias et al. 2009) to the effect that the WELAS instrument was not good at resolving "the largest fog droplets".

Getting two optical sizing instruments to agree is always problematic, and I feel the discrepancies need a more careful discussion in this paper. Are there any other independent measurements that could help resolve this issue, and provide some better evidence that the WELAS underestimates the number of droplets above 8 µm and the FM-100 underestimates them below this size?

The scatterplots shown in Figure 1 are an attempt to provide some evidence for this choice, but an independent source of information would be better. In fact, the data shown in Figure 1 makes me wonder even more about the agreement between the two instruments. The data for the size range in which the instruments should be most comparable (6-8 µm) does have a best fit slope that is close to 1:1. However, looking at the data I would guess that the slope is determined by the few very high values of number concentration. There is considerable spread in the data for all the size ranges, which doesn't exactly

inspire great confidence that the instruments are measuring the same thing. Figure 2 gives values for the correlation coefficient (expressed as **ρ**). If I am interpreting this correctly, this means that even in the two best cases (4-6 um and 6-8 um), the variance in the FSSP measurements explained by the WELAS was 40% and 30% respectively. That low level of agreement isn't all that comforting.

Given that most of the conclusions drawn as a result of these measurements depend on having a believable droplet size distribution, I feel that more discussion about the discrepancies between the WELAS and the FSSP are warranted.

2)      Page 4, line 25

The authors assume state that they measure dry aerosol particle size distributions at a relative humidity of "less than 50%". Unfortunately, 50% relative humidity doesn't mean that the particles are dry. Once wetted, ammonium sulfate (as one example) does not effloresce until below 40% relative humidity (Tang, et al. Aerosol Sci. Technol. 23, 443-453, 1995). Similar observations have been made for other compounds (e.g., Zieger, P., et al., Influence of water uptake on the aerosol particle light scattering coefficients of the Central European aerosol, *Tellus B; Vol 66, 2014*.) Assuming particles are dry at these relative humidities can lead to an overestimation of the solute mass. Further discussion and evaluation of how this potential overestimation may influence the conclusions drawn based of the measurements is necessary.

3)      Page 7, line 24

The authors assume an internal mixture when predicting the CCN number. They cite a paper (Jurányi et al. 2013) that indicates the aerosol at the site is externally mixed, but that this doesn't significantly influence the calculation of CCN. On the other hand, (Hallberg, Ogren et al. 1992, Hallberg, Ogren et al. 1994) show that black carbon and sulfate aerosols of the same size activate to different extents in cloud and polluted fogs. While clear, the differences in scavenging measured in the Hallberg et al. papers may not be enough to greatly change the CCN concentration, since there is plenty of unactivated aerosol left in the cloud. It would be nice, however, to have a bit more detailed discussion of the possible effects of external mixing on the estimation of CCN concentrations.

Along these lines, a plot of the scavenged fraction of the Nd/Naerosol ratio would be nice to give the reader a feel for the general scavenging fraction at this site. Figure 3 provides one version of this ratio, but time series plots would be nice too.

**Minor comments, questions**

1)      Page 4, line 2

I'm not sure what is meant by the sentence "these distributions overlap with each other at a diameter which fluctuates between 5 to 9 um". From the previous page the overlap region between the WELAS and the FSSP is given as 2-40um. It appears from Figure 1 that the meaning is that the two instruments typically give similar values in the interval between 6-8 um. Is this the case? If so, why is there a difference in the interval – 5-9 (in the text) vs. 6-8 (in the figure caption)?

2)      Page 5, line 32

"scatter svery large" should be "scatter is very large"

3)      Page 13, line 12

I'm not sure this "classical" description of growing, thermodynamically activated droplets limiting supersaturation in the fogs is appropriate. This has been pretty much disproven by all the evidence presented previously in the paper – the fact that there are so many hydrated but unactivated particles taking up water in these fogs that it isn't necessarily the thermodynamically activated ones that are responsible for determining the peak supersaturation.